# Promoting Coordination through Policy Regularization in Multi-Agent Deep Reinforcement Learning

## Abstract

A central challenge in multi-agent reinforcement learning is the induction of coordination between agents of a team. In this work, we investigate how to promote inter-agent coordination using policy regularization and discuss two possible avenues respectively based on inter-agent modelling and synchronized sub-policy selection. We test each approach in four challenging continuous control tasks with sparse rewards and compare them against three baselines including MADDPG, a state-of-the-art multi-agent reinforcement learning algorithm. To ensure a fair comparison, we rely on a thorough hyper-parameter selection and training methodology that allows a fixed hyper-parameter search budget for each algorithm and environment. We consequently assess both the hyper-parameter sensitivity, sample-efficiency and asymptotic performance of each learning method. Our experiments show that the proposed methods lead to significant improvements on cooperative problems. We further analyse the effects of the proposed regularizations on the behaviors learned by the agents.

## 1    Introduction

Multi-Agent Reinforcement Learning (MARL) refers to the task of training an agent to maximize its expected return by interacting with an environment that contains other learning agents. It represents a challenging branch of Reinforcement Learning (RL) with interesting developments in recent years (Hernandez-Leal et al., 2018). A popular framework for MARL is the use of a Centralized Training and a Decentralized Execution (CTDE) procedure (Lowe et al., 2017; Foerster et al., 2018; Iqbal & Sha, 2019; Foerster et al., 2019; Rashid et al., 2018). It is typically implemented by training critics that approximate the value of the joint observations and actions, which are used to train actors restricted to the observation of a single agent. Such critics, if exposed to coordinated joint actions leading to high returns, can steer the agents' policies toward these highly rewarding behaviors. However, these approaches depend on the agents luckily stumbling on these actions in order to grasp their benefit. Thus, it might fail in scenarios where coordination is unlikely to occur by chance. We hypothesize that in such scenarios, coordination-promoting inductive biases on the policy search could help discover coordinated behaviors more efficiently and supersede task-specific reward shaping and curriculum learning.

In this work, we explore two different priors for successful coordination and use these to regularize the learned policies. The first avenue, TeamReg, assumes that an agent must be able to predict the behavior of its teammates in order to coordinate with them. The second, CoachReg, supposes that coordinating agents individually recognize different situations and synchronously use different sub-policies to react to them. In the following sections we show how to derive practical regularization terms from these premises and meticulously evaluate them[1].

Our contributions are twofold. First, we propose two novel approaches that aim at promoting coordination in multi-agent systems. Our methods augment CTDE MARL algorithms with additional multi-agent objectives that act as regularizers and are optimized jointly with the main return-maximization objective. Second, we design two new sparse-reward cooperative tasks in the

---

[1]Source code for the algorithms and environments will be made public upon publication of this work.

multi-agent particle environment (Mordatch & Abbeel, 2018). We use them along with two standard multi-agent tasks to present a detailed evaluation of our approaches against three different baselines. Finally, we validate our methods' key components by performing an ablation study. Our experiments suggest that our TeamReg objective provides a dense learning signal that helps to guide the policy towards coordination in the absence of external reward, eventually leading it to the discovery of high performing team strategies in a number of cooperative tasks. Similarly, by enforcing synchronous sub-policy selections, CoachReg enables to fine-tune a sub-behavior for each recognized situation yielding significant improvements on the overall performance.

## 2 BACKGROUND

### 2.1 MARKOV GAMES

In this work we consider the framework of Markov Games (Littman, 1994), a multi-agent extension of Markov Decision Processes (MDPs) with $N$ independent agents. A Markov Game is defined by the tuple $\langle \mathcal{S}, \mathcal{T}, \mathcal{P}, \{\mathcal{O}^i, \mathcal{A}^i, \mathcal{R}^i\}_{i=1}^N \rangle$. $\mathcal{S}$, $\mathcal{T}$, and $\mathcal{P}$ respectively are the set of all possible states, the transition function and the initial state distribution. While these are global properties of the environment, $\mathcal{O}^i$, $\mathcal{A}^i$ and $\mathcal{R}^i$ are individually defined for each agent $i$. They are respectively the observation functions, the sets of all possible actions and the reward functions. At each time-step $t$, the global state of the environment is given by $s_t \in \mathcal{S}$ and every agent's individual action vector is denoted by $a_t^i \in \mathcal{A}^i$. To select their action, each agent $i$ has only access to its own observation vector $o_t^i$ which is extracted by its observation function $\mathcal{O}^i$ from the global state $s_t$. The initial global state $s_0$ is sampled from the initial state distribution $\mathcal{P} : \mathcal{S} \to [0, 1]$ and the next states of the environment $s_{t+1}$ are sampled from the probability distribution over the possible next states given by the transition function $\mathcal{T} : \mathcal{S} \times \mathcal{S} \times \mathcal{A}^1 \times ... \times \mathcal{A}^N \to [0, 1]$. Finally, at each time-step, each agent receives an individual scalar reward $r_t^i$ from its reward function $\mathcal{R}^i : \mathcal{S} \times \mathcal{S} \times \mathcal{A}^1 \times ... \times \mathcal{A}^N \to \mathbb{R}$. Agents aim at maximizing their expected discounted return $\mathbb{E}\left[\sum_{t=0}^T \gamma^t r_t^i\right]$ over the time horizon $T$, where $\gamma \in [0, 1]$ is a discount factor.

### 2.2 MULTI-AGENT DEEP DETERMINISTIC POLICY GRADIENT (MADDPG)

MADDPG (Lowe et al., 2017) is an adaptation of the Deep Deterministic Policy Gradient algorithm (DDPG) (Lillicrap et al., 2015) to the multi-agent setting. It allows the training of cooperating and competing decentralized policies through the use of a centralized training procedure. In this framework, each agent $i$ possesses its own deterministic policy $\mu^i$ for action selection and critic $Q^i$ for state-action value estimation, which are respectively parametrized by $\theta^i$ and $\phi^i$. All parametric models are trained off-policy from previous transitions $\zeta_t := (\mathbf{o}_t, \mathbf{a}_t, \mathbf{r}_t, \mathbf{o}_{t+1})$ uniformly sampled from a replay buffer $\mathcal{D}$. Note that $\mathbf{o}_t := [o_t^1, ..., o_t^N]$ is the joint observation vector and $\mathbf{a}_t := [a_t^1, ..., a_t^N]$ is the joint action vector, obtained by concatenating the individual observation vectors $o_t^i$ and action vectors $a_t^i$ of all $N$ agents. Each centralized critic is trained to estimate the expected return for a particular agent $i$ using the Deep Q-Network (DQN) (Mnih et al., 2015) loss:

$$\mathcal{L}^i(\phi^i) = \mathbb{E}_{\zeta_t \sim \mathcal{D}}\left[ \frac{1}{2}\left(Q^i(\mathbf{o}_t, \mathbf{a}_t; \phi^i) - (r_t^i + \gamma Q^i(\mathbf{o}_{t+1}, \mathbf{a}_{t+1}; \bar{\phi}^i))\right)^2 \Big|_{a_{t+1}^j = \mu^j(o_{t+1}^j; \bar{\theta}^j) \, \forall j} \right] \quad (1)$$

For a given set of weights $w$, we define its target counterpart $\bar{w}$, updated from $\bar{w} \leftarrow \tau w + (1 - \tau)\bar{w}$ where $\tau$ is a hyper-parameter. Each policy is updated to maximize the expected discounted return of the corresponding agent $i$ :

$$J_{PG}^i(\theta^i) = \mathbb{E}_{\mathbf{o}_t \sim \mathcal{D}}\left[ Q^i(\mathbf{o}_t, \mathbf{a}_t)\big|_{a_t^i = \mu^i(o_t^i; \theta^i), \, a_t^j = \mu^j(o_t^j; \bar{\theta}^j) \, \forall j \neq i} \right]$$

$$\nabla_{\theta^i} J_{PG}^i(\theta^i) = \mathbb{E}_{\mathbf{o}_t \sim \mathcal{D}}\left[ \nabla_{\theta^i}\mu^i(o_t^i; \theta_i)\nabla_{a_t^i} Q^i(\mathbf{o}_t, \mathbf{a}_t)\big|_{a_t^i = \mu^i(o_t^i; \theta^i), \, a_t^j = \mu^j(o_t^j; \bar{\theta}^j) \, \forall j \neq i} \right]$$

$$(2)$$

By taking into account *all* agents' observation-action pairs when guiding an agent's policy, the value-functions are trained in a centralized, stationary environment, despite taking place in a multi-agent setting. In addition, this mechanism can allow to implicitly learn coordinated strategies that can then be deployed in a decentralized way. However, this procedure does not encourage the *discovery* of coordinated strategies since high-reward behaviors have to be randomly experienced through unguided exploration. This work aims at alleviating this limitation.

# 3    RELATED WORK

Many works in MARL consider explicit communication channels between the agents and distinguish between communicative actions (e.g. broadcasting a given message) and physical actions (e.g. moving in a given direction) (Foerster et al., 2016; Mordatch & Abbeel, 2018; Lazaridou et al., 2016). Consequently, they often focus on the emergence of language, considering tasks where the agents must discover a common communication protocol in order to succeed. Deriving a successful communication protocol can already be seen as coordination in the communicative action space and can enable, to some extent, successful coordination in the physical action space (Ahilan & Dayan, 2019). Yet, explicit communication is not a necessary condition for coordination as agents can rely on physical communication (Mordatch & Abbeel, 2018; Gupta et al., 2017).

Approaches to shape RL agents' behaviors with respect to other agents have also been explored. Strouse et al. (2018) use the mutual information between the agent's policy and a goal-independent policy to shape the agent's behavior towards hiding or spelling out its current goal. However, this approach is only applicable for tasks with an explicit goal representation and is not specifically intended for coordination. Jaques et al. (2019) approximate the direct causal effect between agent's actions and use it as an intrinsic reward to encourage social empowerment. This approximation relies on each agent learning a model of other agents' policies to predict its effect on them. In general, this type of behavior prediction can be referred to as *agent modelling* (or opponent modelling) and has been used in previous work to enrich representations (Hernandez-Leal et al., 2019), to stabilise the learning dynamics (He et al., 2016) or to classify the opponent's play style (Schadd et al., 2007). In our work, agent modelling is extended to derive a novel incentive toward team-predictable behaviors.

Finally, Barton et al. (2018) propose convergent cross mapping (CCM) to measure the degree of effective coordination between two agents. Although this may represent an interesting avenue for behavior analysis, it fails to provide a tool for effectively enforcing coordination as CCM must be computed over long time series which makes it an impractical learning signal for single-step temporal difference methods. In this work, we design two coordination-driven multi-agent approaches that do not rely on the existence of explicit communication channels and allow to carry the learned coordinated behaviors at test time, when all agents act in a decentralized fashion.

# 4    COORDINATION AND POLICY REGULARIZATION

Intuitively, coordination can be defined as an agent's behavior being informed by the one of another agent, i.e. structure in the agents' interactions. Namely, a team where agents act independently of one another would not be coordinated. To promote such structure, our proposed methods rely on team-objectives as regularizers of the common policy gradient update. In this regard, our approach is closely related to General Value Functions and Auxiliary tasks (Sutton & Barto, 2018) used in Deep RL to learn efficient representations (Jaderberg et al., 2019). However, this work's novelty lies in the explicit bias of agents' policy towards either predictability for their teammates or synchronous sub-policy selection. Pseudocodes of our implementations are provided in Appendix C (see Algorithms 1 and 2).

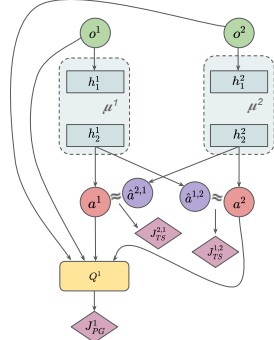

Figure 1: Illustration of TeamReg with two agents. Each agent's policy is equipped with additional heads that are trained to predict other agents' actions and every agent is regularized to produce actions that its teammates correctly predict. Note that the method is depicted for agent 1 only to avoid cluttering.

## 4.1    TEAM REGULARIZATION

The structure of coordinated interactions can be leveraged to attain a certain degree of predictability of one agent's behavior with respect to its teammate(s). We hypothesize that the reciprocal also holds i.e. that promoting agents' predictability could foster such team structure and lead to more coordinated behaviors. This assumption is cast into the decentralized framework by training agents to predict their teammates' actions given only their own observation. For continuous control, the loss is defined as the Mean Squared Error (MSE) between the predicted

and true actions of the teammates, yielding a teammate-modelling secondary objective. While the previous work of Hernandez-Leal et al. (2019) focus on stationary, non-learning teammates and exclusively use this approach to learn richer internal representations, we propose to extend this objective to drive the teammates' behaviors closer to the prediction by leveraging a differentiable action selection mechanism. We call *team-spirit* this novel objective $J_{TS}^{i,j}$ between agents $i$ and $j$:

$$J_{TS}^{i,j}(\theta^i, \theta^j) = \mathbb{E}_{\mathbf{o}_t \sim \mathcal{D}} \left[ -\text{MSE}(\hat{a}_t^{i,j}, a_t^j) \Big|_{a_t^j = \mu^j(o_t^j), \hat{a}_t^{i,j} = \hat{\mu}^{i,j}(o_t^i)} \right] \tag{3}$$

$$= -\mathbb{E}_{\mathbf{o}_t \sim \mathcal{D}} \left[ \frac{1}{2} \sum_{k=1}^{|\mathcal{A}^j|} (\hat{a}_{t,k}^{i,j} - a_{t,k}^j)^2 \Big|_{a_t^j = \mu^j(o_t^j), \hat{a}_t^{i,j} = \hat{\mu}^{i,j}(o_t^i)} \right] \tag{4}$$

where $\hat{\mu}^{i,j}$ is the policy head of agent $i$ trying to predict the action of agent $j$. The total gradient for a given agent $i$ becomes:

$$\nabla_{\theta^i} J_{total}^i(\theta^i) = \nabla_{\theta^i} J_{PG}^i(\theta^i) + \lambda_1 \sum_j \nabla_{\theta^i} J_{TS}^{i,j}(\theta^i, \theta^j) + \lambda_2 \sum_j \nabla_{\theta^i} J_{TS}^{j,i}(\theta^j, \theta^i) \tag{5}$$

where $\lambda_1$ and $\lambda_2$ are hyper-parameters that respectively weight how well an agent should predict its teammates' actions, and how predictable an agent should be for its teammates. We call TeamReg this dual regularization from team-spirit objectives. Figure 1 summarizes these interactions.

## 4.2 COACH REGULARIZATION

In order to foster structured agents interactions, this method aims at teaching the agents to recognize different situations and synchronously select corresponding sub-behaviors.

### 4.2.1 SUB-POLICY SELECTION

Firstly, to enable explicit sub-behavior selection, we propose *policy masks* that modulate the agents' policy. A policy mask $u^j$ is a one-hot vector of size $K$ with its $j^{\text{th}}$ component set to one. In practice, we use policy masks to perform dropout (Srivastava et al., 2014) in a structured manner on $\tilde{h}_1 \in \mathbb{R}^M$, the pre-activations of the first hidden layer $h_1$ of the policy network $\pi$. To do so, we construct the vector $\boldsymbol{u}^j$, which is the concatenation of $C$ copies of $u^j$, in order to reach the dimensionality $M = C * K$. The element-wise product $\boldsymbol{u}^j \odot \tilde{h}_1$ is then performed and only the units of $\tilde{h}_1$ at indices $m$ modulo $K = j$ are kept for $m = 0, \ldots, M-1$. In our contribution, each agent $i$ generates $e_t^i$, its own policy mask, from its observation $o_t^i$. Here, a simple linear layer $l^i$ is used to produce a categorical probability distribution $p^i(e_t^i | o_t^i)$ from which the one-hot vector is sampled:

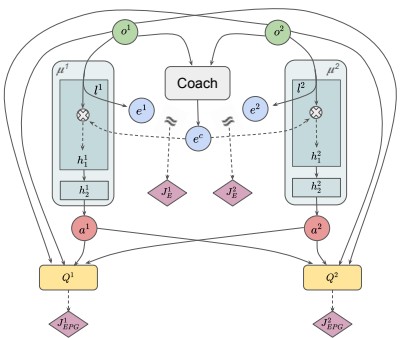

Figure 2: Illustration of CoachReg with two agents. A central model, the coach, takes all agents' observations as input and outputs the current mode (policy mask). Agents are regularized to predict the same mask from their local observations only and optimize the corresponding sub-policy.

$$p^i(e_t^i = u^j | o_t^i) = \text{softmax}(l^i(o_t^i; \theta^i))_j = \frac{\exp\left(l^i(o_t^i; \theta^i)_j\right)}{\sum_{k=0}^{K-1} \exp\left(l^i(o_t^i; \theta^i)_k\right)} \tag{6}$$

To our knowledge, while this method draws similarity to the options and hierarchical frameworks (Sutton & Barto, 2018; Ahilan & Dayan, 2019) and to policy dropout for exploration (Xie et al., 2018), it is the first to introduce an agent induced modulation of the policy network by a structured dropout that is decentralized at evaluation and without an explicit communication channel. Although the policy masking mechanism enables the agent to swiftly switch between sub-policies it does not encourage the agents to synchronously modulate their behavior.

### 4.2.2 SYNCHRONOUS SUB-POLICY SELECTION

To promote synchronization we introduce the *coach* entity, parametrized by $\psi$, which learns to produce policy-masks $e_t^c$ from the joint observations, i.e. $p^c(e_c^i | \mathbf{o}_t; \psi)$. The coach is used at training

time only and drives the agents toward synchronously selecting the same behavior mask. In other words, the coach is trained to output masks that (1) yield high returns when used by the agents and (2) are predictable by the agents. Similarly, each agent is regularized so that (1) its private mask matches the coach's mask and (2) it derives efficient behavior when using the coach's mask. At evaluation time, the coach is removed and the agents only rely on their own policy masks. The policy gradient loss when agent $i$ is provided with the coach's mask is given by:

$$J^i_{EPG}(\theta^i, \psi) = \mathbb{E}_{\mathbf{o}_t, \mathbf{a}_t \sim \mathcal{D}} \left[ Q^i(\mathbf{o}_t, \mathbf{a}_t) \big|_{a^i_t = \mu(o^i_t, e_t; \theta^i),\, e_t \sim p^c(\cdot | \mathbf{o}_t; \psi)} \right] \qquad (7)$$

The difference between the mask of agent $i$ and the coach's one is measured from the Kullback–Leibler divergence:

$$J^i_E(\theta^i, \psi) = \mathbb{E}_{\mathbf{o}_t \sim \mathcal{D}} \left[ D_{KL} \left( p^c(\cdot | \mathbf{o}_t; \psi) \| p^i(\cdot | o^i_t; \theta^i) \right) \right] \qquad (8)$$

The total gradient for agent $i$ is:

$$\nabla_{\theta^i} J^i_{total}(\theta^i) = \nabla_{\theta^i} J^i_{PG}(\theta^i) + \lambda_1 \nabla_{\theta^i} J^i_E(\theta^i, \psi) + \lambda_2 \nabla_{\theta^i} J^i_{EPG}(\theta^i, \psi)$$

$$\nabla_{\theta^i} J^i_{EPG}(\theta^i, \psi) = \mathbb{E}_{\mathbf{o}_t, \mathbf{a}_t \sim \mathcal{D}} \left[ \nabla_{\theta^i} \mu(o^i_t, e_t; \theta^i) \nabla_{a^i_t} Q^i(\mathbf{o}_t, \mathbf{a}_t) \big|_{a^i_t = \mu(o^i_t, e_t),\, e_t \sim p^c(\cdot | \mathbf{o}_t; \psi)} \right] \qquad (9)$$

with $\lambda_1$ and $\lambda_2$ the regularization coefficients. Similarly, the coach is trained with the following dual objective, weighted by the $\lambda_3$ coefficient:

$$\nabla_\psi J^c_{total}(\psi) = \frac{1}{N} \sum_{i=1}^{N} \left( \nabla_\psi J^i_{EPG}(\theta^i, \psi) + \lambda_3 \nabla_\psi J^i_E(\theta^i, \psi) \right) \qquad (10)$$

In order to propagate gradients through the sampled policy mask we reparametrized the categorical distribution using the Gumbel-softmax trick (Jang et al., 2017) with a temperature of 1. We call this coordinated sub-policy selection regularization CoachReg and illustrate it in Figure 2.

## 5    TRAINING ENVIRONMENTS

All of our tasks are based on the OpenAI multi-agent particle environments (Mordatch & Abbeel, 2018). SPREAD and CHASE were introduced by (Lowe et al., 2017). We use SPREAD as is but with sparse rewards only. CHASE is modified with a prey controlled by repulsion forces and only the predators are learnable, as we wish to focus on coordination in cooperative tasks. Finally we introduce COMPROMISE and BOUNCE where agents are explicitly tied together. While non-zero return can be achieved in these tasks by selfish agents, they all benefit from coordinated strategies and optimal return can only be achieved by agents working closely together. Figure 3 presents visualizations and a brief description of all four tasks. A detailed description is provided in Appendix A. In all tasks, agents receive as observation their own global position and velocity as well as the relative position of other entities. Note that work showcasing experiments on this environment often use discrete action spaces and (dense) reward shaping (e.g. the proximity with the objective) (Iqbal & Sha, 2019; Lowe et al., 2017; Jiang & Lu, 2018). However, in our experiments, agents learn with continuous action spaces and from sparse rewards.

## 6    RESULTS AND DISCUSSION

The proposed methods offer a way to incorporate new inductive biases in CTDE multi-agent policy search algorithms. In this work, we evaluate them by extending MADDPG, a state of the art algorithm widely used in the MARL litterature. We compare against vanilla MADDPG as well as two of its variations in the four cooperative multi-agent tasks described in Section 5. The first variation (DDPG) is the single-agent counterpart of MADDPG (decentralized training). The second (MADDPG + sharing) shares the policy and value-function models across agents.

To offer a fair comparison between all methods, the hyper-parameter search routine is the same for each algorithm and environment (see Appendix D.1). For each search-experiment (one per algorithm per environment), 50 randomly sampled hyper-parameter configurations each using 3 training seeds (total of 150 runs) are used to train the models for $15,000$ episodes. For each algorithm-environment pair, we then select the best hyper-parameter configuration for the final comparison and retrain them on 10 seeds for twice as long. We give more details about the training setup and model selection in Appendix B and D.2. The results of the hyperparameter searches are given in Appendix D.5.

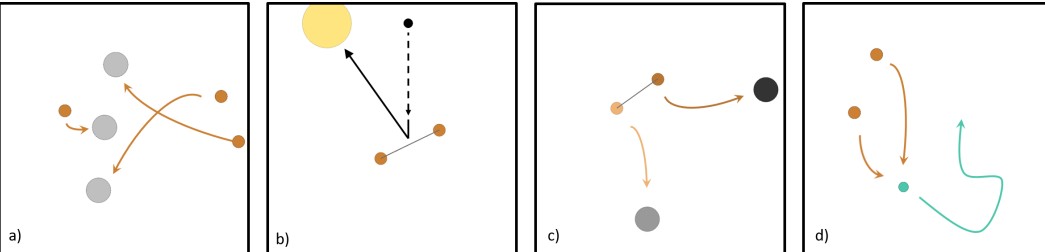

Figure 3: Multi-agent tasks used in this work. (a) SPREAD: Agents must spread out and cover a set of landmarks. (b) BOUNCE: Two agents are linked together by a spring and must position themselves so that the falling black ball bounces towards a target. (c) COMPROMISE: Two linked agents must compete or cooperate to reach their own assigned landmark. (d) CHASE: Two agents chase a (non-learning) prey (turquoise) that moves w.r.t repulsion forces from predators and walls.

## 6.1 ASYMPTOTIC PERFORMANCE

From the average learning curves reported in Figure 4 we observe that CoachReg significantly improves performance on three environments (SPREAD, BOUNCE and COMPROMISE) and performs on par with the baselines on the last one (CHASE). The same can be said for TeamReg, except on COMPROMISE, the only task with an adversarial component, where it significantly underperforms compared to the other algorithms. We discuss this specific case in Section 6.3. Finally, parameter sharing is the best performing choice on CHASE, yet this superiority is restricted to this task where the optimal play is to move symmetrically and squeeze the prey into a corner.

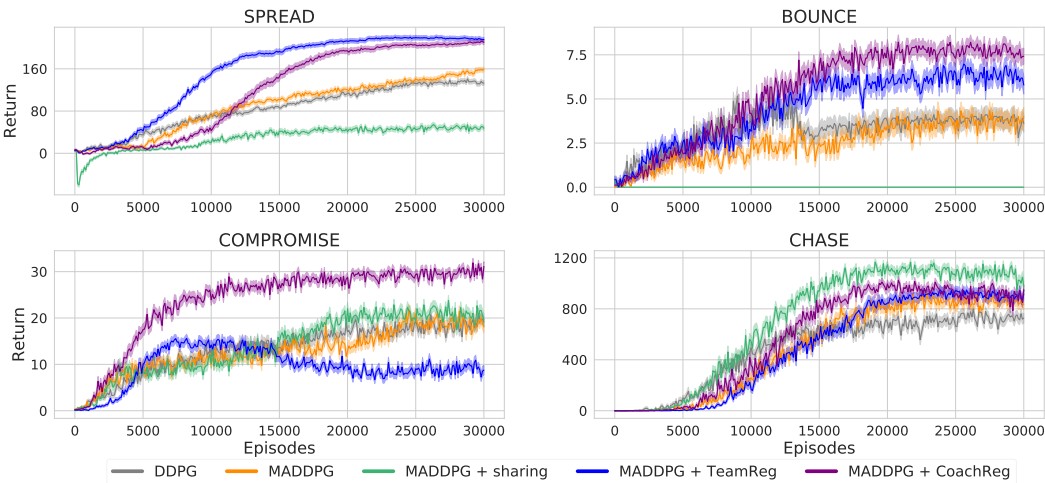

Figure 4: Learning curves (mean return over agents) for all algorithms on all four environments. Solid lines are the mean and envelopes are the Standard Error (SE) across the 10 training seeds.

## 6.2 ABLATION STUDY

Additionally to our two proposed algorithms and the three baselines, we present results for two ablated versions of our methods. The first ablation (MADDPG + agent modelling) is similar to TeamReg but with $\lambda_2 = 0$, which results in only enforcing agent modelling (i.e. agent predictability is not encouraged). The second ablation (MADDPG + policy mask) is structurally equivalent to CoachReg, but with $\lambda_{1,2,3} = 0$, which means that agents still predict and apply a mask to their own policy, but synchronicity is not encouraged. Figure 12 and 13 (Appendix D.6) present the results of the corresponding hyper-parameter search and Figure 5 shows the learning curves for our full regularization approaches, their respective ablated versions and MADDPG.

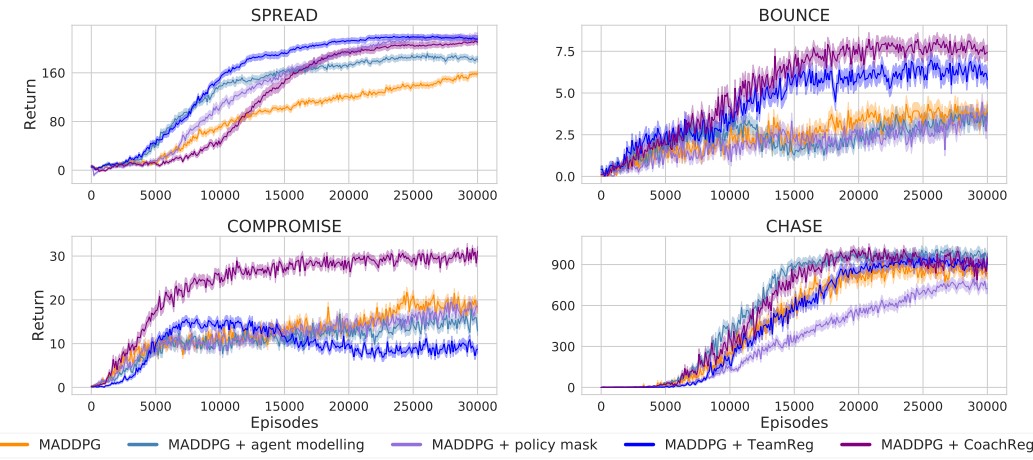

Figure 5: Learning curves (mean return over agents) for the ablated algorithms on all environments. Solid lines are the mean and envelopes are the Standard Error (SE) across the 10 training seeds.

The use of unsynchronized policy masks might result in swift and unpredictable behavioral changes and make it difficult for agents to perform together and coordinate. Experimentally, "MADDPG + policy mask" performs similarly or worse than MADDPG on all but one environment, and never outperforms the full CoachReg approach. However, policy masks alone seem enough to succeed on SPREAD, which is about selecting a landmark from a set. Regarding "MADDPG + agent modelling", it does not drastically improve on MADDPG apart from on the SPREAD environment, and the full TeamReg approach shows improvement over its ablated version except on the COMPROMISE task, which we discuss in Section 6.3.

### 6.3 EFFECTS OF ENFORCING PREDICTABLE BEHAVIOR

First, we investigate the reason for TeamReg's poor performance on COMPROMISE. Then, we analyse how TeamReg might be helpful in other environments.

COMPROMISE is the only task with a competitive component (and the only one in which agents do not share their rewards). The two agents being linked, a good policy has both agents reach their landmark successively (maybe by simply having both agents navigate towards the closest landmark). However, if one agent never reaches for its landmark, the optimal strategy for the other one becomes to drag it around and always go for its own, leading to a strong imbalance in the return cumulated by both agents. While this scenario very rarely occurs for the other algorithms, we found TeamReg to often lead to such domination cases (see Figure 14 in Appendix E). Figure 6 depicts the agents' performance difference for every 150 runs of the hyperparameter search for TeamReg and the baselines, and shows that (1) TeamReg is the only algorithm that does lead to large imbalances in performance between the two agents and (2) that these cases where one agent becomes dominant are all associated with high values of $\lambda_2$, which drives the agents to behave in a predictable

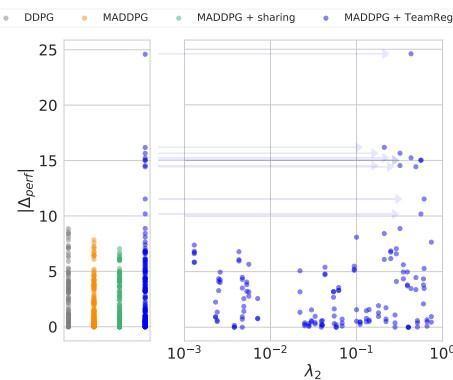

Figure 6: Average performance difference ($\Delta_{perf}$) between the two agents in COM-PROMISE for each 150 runs of the hyper-parameter searches (left). All occurrences of abnormally high performance difference are associated with high values of $\lambda_2$ (right).

fashion to one another. However, the dominated agent eventually gets exposed more and more to sparse reward gathered by being dragged (by chance) onto its own landmark, picks up the goal of the task and starts pulling in its own direction, which causes the average return over agents to drop as we see in Figure 4. This experiment demonstrates that using a predictability-based team-regularization

in a competitive task can be harmful; quite understandably, you might not want to optimize an objective that aims at making your behavior predictable to your opponent.

On SPREAD and BOUNCE, TeamReg significantly improves the performance over the baselines. We aim to analyze here the effects of $\lambda_2$ on cooperative tasks and investigate if it does make the agent modelling task more successful (by encouraging the agent to be predictable). To this end, we compare the best performing hyper-parameter configuration for TeamReg on the SPREAD environment with its ablated versions. The average return and team-spirit loss defined in Section 4.1 are presented in Figure 7 for these three experiments.

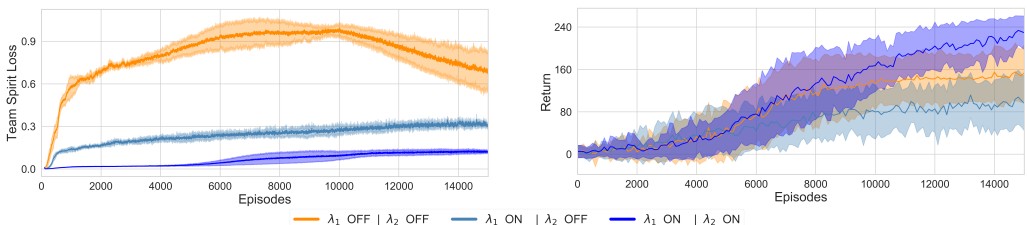

Figure 7: Comparison between enabling and disabling the regularizing weights $\lambda_1$ and $\lambda_2$ for MADDPG+TeamReg on the SPREAD environment. Values are averaged over the 3 agents and over the 3 seeds used in the hyper-parameter exploration.

Initially, due to the weight initialization, the predicted and actual actions both have relatively small norms yielding small values of team-spirit loss. As training goes on ($\sim$1000 episodes), the norms of the action-vector increase and the regularization loss becomes more important. As expected, $\lambda_2 OFF | \lambda_1 OFF$ leads to the highest team-spirit loss as it is not trained to predict the actions of other agents correctly. When using only the agent-modelling objective ($\lambda_1 ON$), the agents significantly decrease the team-spirit loss, but it never reaches values as low as when using the full TeamReg objective. Finally, when also pushing agents to be predictable ($\lambda_2 ON$), the agents best predict each others' actions and performance is also improved. We also notice that the team-spirit loss increases when performance starts to improve i.e. when agents start to master the task ($\sim$8000 episodes). Indeed, once the reward maximisation signals becomes stronger, the relative importance of the second task is reduced. We hypothesize that being predictable with respect to one-another may push agents to explore in a more structured and informed manner in the absence of reward signal, as similarly pursued by intrinsic motivation approaches (Chentanez et al., 2005).

## 6.4 ANALYSIS OF SYNCHRONOUS SUB-POLICY SELECTION

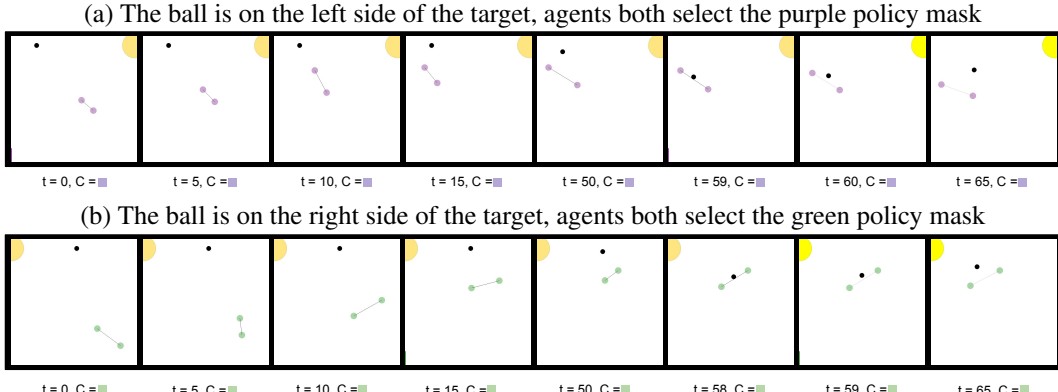

Figure 8: Visualization of two different BOUNCE evaluation episodes. Note that here, the agents' colors represent their chosen policy mask. Agents have learned to synchronously identify two distinct situations and act accordingly. The coach's masks (not used at evaluation time) are displayed with the timestep at the bottom of each frame.

In this section we aim at experimentally verifying that CoachReg yields the desired behavior: agents *synchronously* alternating between *varied* sub-policies. A special attention is given when the sub-policies are interpretable. To this end we record and analyze the agents' policy masks on 100 different episodes for each task.

From the collected masks, we reconstructed the empirical mask distribution of each agent (see Figure 15 in Appendix F.1) whose entropy provides an indication of the mask diversity used by a given agent. Figure 9 (a) shows the mean entropy for each environment compared to the entropy of Categorical Uniform Distributions of size $k$ ($k$-CUD). It shows that, on all the environments, agents use at least two distinct masks by having non-zero entropy. In addition, agents tend to alternate between masks with more variety (close to uniformly switching between 3 masks) on SPREAD (where there are 3 agents and 3 goals) than on the other environments (comprised of 2 agents). To test if agents are synchronously selecting the same policy mask at test time (without a coach), we compute the Hamming proximity between the agents' mask sequences with $1 - D_h$ where $D_h$ is the Hamming distance, i.e. the number of timesteps where the two sequences are different divided by the total number of timesteps. From Figure 9 (b) we observe that agents are producing similar mask sequences. Notably, their mask sequences are significantly more similar that the ones of two agent randomly choosing between two masks at each timestep. Finally, we observe that some settings result in the agents coming up with interesting strategies, like the one depicted in Figure 8 where the agents alternate between two sub-policies depending on the position of the target. More cases where the agents change sub-policies during an episode are presented in Appendix F.1. These results indicate that, in addition to improving the performance on coordination tasks, CoachReg indeed yields the expected behaviors. An interesting following work would be to use entropy regularization to increase the mask usage variety and mutual information to further disentangle sub-policies.

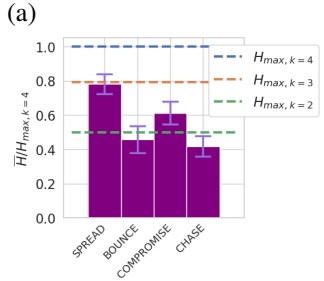

(a)

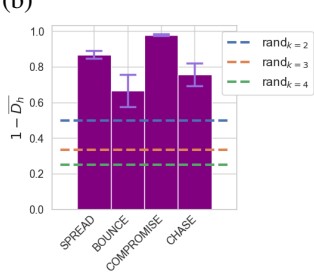

(b)

Figure 9: (a) Entropy of the policy mask distributions for each task, averaged over agents and training seeds. $H_{max,k}$ is the entropy of a $k$-CUD. (b) Hamming Proximity between the policy mask sequence of each agent averaged across agent pairs and seeds. $\text{rand}_k$ stands for agents independently sampling their masks from $k$-CUD. Error bars are SE across seeds.

## 6.5 ROBUSTNESS TO HYPER-PARAMETERS

Stability across hyper-parameter configurations is a recurring challenge in Deep RL. The average performance for each sampled configuration allow to empirically evaluate the robustness of an algorithm w.r.t. its hyper-parameters. We share the full results of the hyper-parameter searches in Figures 10, 11, 12 and 13 in Appendix D.5 and D.6. Figure 11 shows that while most algorithms can perform reasonably well with the correct configuration, our proposed coordination regularizers can improve robustness to hyper-parameter despite the fact that they have more hyper-parameters to search over. Such robustness can be of great value with limited computational budgets.

## 6.6 ROBUSTNESS TO THE NUMBER OF AGENTS

To assess how the proposed methods perform when using an greater number of agents, we present additional experiments for which the number of agents in the SPREAD task is gradually increased from three to six agents. The results presented in Figure 18 (Appendix G) show that the performance benefits provided by our methods hold when the number of agents is increased. Strikingly, we also note how quickly the performance of all methods drop when the number of agents rises. Indeed, with each new agent, the coordination problem becomes more and more difficult, and that might explain why our methods that promote coordination maintain a higher degree of performance in the case of 4 agents. Nonetheless, estimating the value function also becomes increasingly challenging as the input space grows exponentially with the number of agents. In the sparse reward setting, the complexity of the task soon becomes too difficult and none of the algorithms is able to solve it with six agents.

# 7 CONCLUSION

In this work we introduced two policy regularization methods to promote multi-agent coordination within the CTDE framework: TeamReg, which is based on inter-agent action predictability and CoachReg that relies on synchronized behavior selection. A thorough empirical evaluation of these methods showed that they significantly improve asymptotic performances on cooperative multi-agent tasks. Interesting avenues for future work would be to study the proposed regularizations on other policy search methods as well as to combine both incentives and investigate how the two coordinating objectives interact. Finally, a limitation of the current formulation is that it relies on single-step metrics, which simplifies off-policy learning but also limits the longer-term coordination opportunities. A promising direction is thus to explore model-based planning approaches to promote long-term multi-agent interactions.

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

## A    TASKS DESCRIPTIONS

**SPREAD** (Figure 3a): In this environment, there are 3 agents (small orange circles) and 3 landmarks (bigger gray circles). At every timestep, agents receive a team-reward $r_t = n - c$ where $n$ is the number of landmarks occupied by at least one agent and $c$ the number of collisions occurring at that timestep. To maximize their return, agents must therefore spread out and cover all landmarks. Initial agents' and landmarks' positions are random. Termination is triggered when the maximum number of timesteps is reached.

**BOUNCE** (Figure 3b): In this environment, two agents (small orange circles) are linked together with a spring that pulls them toward each other when stretched above its relaxation length. At episode's mid-time a ball (smaller black circle) falls from the top of the environment. Agents must position correctly so as to have the ball bounce on the spring towards the target (bigger beige circle), which turns yellow if the ball's bouncing trajectory passes through it. They receive a team-reward of $r_t = 0.1$ if the ball reflects towards the side walls, $r_t = 0.2$ if the ball reflects towards the top of the environment, and $r_t = 10$ if the ball reflects towards the target. At initialisation, the target's and ball's vertical position is fixed, their horizontal positions are random. Agents' initial positions are also random. Termination is triggered when the ball is bounced by the agents or when the maximum number of timesteps is reached.

**COMPROMISE** (Figure 3c): In this environment, two agents (small orange circles) are linked together with a spring that pulls them toward each other when stretched above its relaxation length. They both have a distinct assigned landmark (light gray circle for light orange agent, dark gray circle for dark orange agent), and receive a reward of $r_t = 10$ when they reach it. Once a landmark is reached by its corresponding agent, the landmark is randomly relocated in the environment. Initial positions of agents and landmark are random. Termination is triggered when the maximum number of timesteps is reached.

**CHASE** (Figure 3d): In this environment, two predators (orange circles) are chasing a prey (turquoise circle). The prey moves with respect to a scripted policy consisting of repulsion forces from the walls and predators. At each timestep, the learning agents (predators) receive a team-reward of $r_t = n$ where $n$ is the number of predators touching the prey. The prey has a greater max speed and acceleration than the predators. Therefore, to maximize their return, the two agents must coordinate in order to squeeze the prey into a corner or a wall and effectively trap it there. Termination is triggered when the maximum number of time steps is reached.

## B    TRAINING DETAILS

In all of our experiments, we use the Adam optimizer (Kingma & Ba, 2014) to perform parameter updates. All models (actors, critics and coach) are parametrized by feedforward networks containing two hidden layers of 128 units. We use the Rectified Linear Unit (ReLU) (Nair & Hinton, 2010) as activation function and layer normalization (Ba et al., 2016) on the pre-activations unit to stabilize the learning. We use a buffer-size of $10^6$ entries and a batch-size of $1024$. We collect 100 transitions by interacting with the environment for each learning update. For all tasks in our hyper-parameter searches, we train the agents for $15,000$ episodes of 100 steps and then re-train the best configuration for each algorithm-environment pair for twice as long ($30,000$ episodes) to ensure full convergence for the final evaluation. The scale of the exploration noise is kept constant for the first half of the training time and then decreases linearly to $0$ until the end of training. We use a discount factor $\gamma$ of $0.95$ and a gradient clipping threshold of $0.5$ in all experiments. Finally for CoachReg, we fixed $K$ to 4 meaning that agents could choose between 4 sub-policies. Since policies' hidden layers are of size 128 the corresponding value for $C$ is 32.

## C  ALGORITHMS

---

**Algorithm 1** Team

---

Randomly initialize $N$ critic networks $Q^i$ and actor networks $\mu^i$
Initialize the target weights
Initialize one replay buffer $\mathcal{D}$
**for** episode from 0 to number of episodes **do**
    Initialize random processes $\mathcal{N}^i$ for action exploration
    Receive initial joint observation $\mathbf{o}_0$
    **for** timestep t from 0 to episode length **do**
        Select action $a_i = \mu^i(o_t^i) + \mathcal{N}_t^i$ for each agent
        Execute joint action $\mathbf{a}_t$ and observe joint reward $\mathbf{r}_t$ and new observation $\mathbf{o}_{t+1}$
        Store transition $(\mathbf{o}_t, \mathbf{a}_t, \mathbf{r}_t, \mathbf{o}_{t+1})$ in $\mathcal{D}$
    **end for**
    Sample a random minibatch of $M$ transitions from $\mathcal{D}$
    **for** each agent $i$ **do**
        Evaluate $\mathcal{L}^i$ and $J_{PG}^i$ from Equations (1) and (2)
        **for** each other agent $(j \neq i)$ **do**
            Evaluate $J_{TS}^{i,j}$ from Equations (3)
            Update actor $j$ with $\theta^j \leftarrow \theta^j + \alpha_\theta \nabla_{\theta^j} \lambda_2 J_{TS}^{i,j}$
        **end for**
        Update critic with $\phi^i \leftarrow \phi^i - \alpha_\phi \nabla_{\phi^i} \mathcal{L}^i$
        Update actor $i$ with $\theta^i \leftarrow \theta^i + \alpha_\theta \nabla_{\theta^i} \left( J_{PG}^i + \lambda_1 \sum_{j=1}^N J_{TS}^{i,j} \right)$
    **end for**
    Update all target weights
**end for**

---

---

**Algorithm 2** Coach

---

Randomly initialize $N$ critic networks $Q^i$, actor networks $\mu^i$ and one coach network $p^c$
Initialize $N$ target networks $Q^{i\prime}$ and $\mu^{i\prime}$
Initialize one replay buffer $\mathcal{D}$
**for** episode from 0 to number of episodes **do**
    Initialize random processes $\mathcal{N}^i$ for action exploration
    Receive initial joint observation $\mathbf{o}_0$
    **for** timestep t from 0 to episode length **do**
        Select action $a_i = \mu^i(o_t^i) + \mathcal{N}_t^i$ for each agent
        Execute joint action $\mathbf{a}_t$ and observe joint reward $\mathbf{r}_t$ and new observation $\mathbf{o}_{t+1}$
        Store transition $(\mathbf{o}_t, \mathbf{a}_t, \mathbf{r}_t, \mathbf{o}_{t+1})$ in $\mathcal{D}$
    **end for**
    Sample a random minibatch of $M$ transitions from $\mathcal{D}$
    **for** each agent $i$ **do**
        Evaluate $\mathcal{L}^i$ and $J_{PG}^i$ from Equations (1) and (2)
        Update critic with $\phi^i \leftarrow \phi^i - \alpha_\phi \nabla_{\phi^i} \mathcal{L}^i$
        Update actor with $\theta^i \leftarrow \theta^i + \alpha_\theta \nabla_{\theta^i} J_{PG}^i$
    **end for**
    **for** each agent $i$ **do**
        Evaluate $J_E^i$ and $J_{EPG}^i$ from Equations (8) and (7)
        Update actor with $\theta^i \leftarrow \theta^i + \alpha_\theta \nabla_{\theta^i} \left( \lambda_1 J_E^i + \lambda_2 J_{EPG}^i \right)$
    **end for**
    Update coach with $\psi \leftarrow \psi + \alpha_\psi \nabla_\psi \frac{1}{N} \sum_{i=1}^N \left( J_{EPG}^i + \lambda_3 J_E^i \right)$
    Update all target weights
**end for**

---

# D HYPER-PARAMETER SEARCH

## D.1 HYPER-PARAMETER SEARCH RANGES

We perform searches over the following hyper-parameters: the learning rate of the actor $\alpha_\theta$, the learning rate of the critic $\omega_\phi$ relative to the actor ($\alpha_\phi = \omega_\phi * \alpha_\theta$), the target-network soft-update parameter $\tau$ and the initial scale of the exploration noise $\eta_{noise}$ for the Ornstein-Uhlenbeck noise generating process (Uhlenbeck & Ornstein, 1930) as used by Lillicrap et al. (2015). When using TeamReg and CoachReg, we additionally search over the regularization weights $\lambda_1$, $\lambda_2$ and $\lambda_3$. The learning rate of the coach is always equal to the actor's learning rate (i.e. $\alpha_\theta = \alpha_\psi$), motivated by their similar architectures and learning signals and in order to reduce the search space. Table 1 shows the ranges from which values for the hyper-parameters are drawn uniformly during the searches.

Table 1: Ranges for hyper-parameter search, the log base is 10

| HYPER-PARAMETER | RANGE |
|---|---|
| $\log(\alpha_\theta)$ | $[-8, -3]$ |
| $\log(\omega_\phi)$ | $[-2, \ \ 2]$ |
| $\log(\tau)$ | $[-3, -1]$ |
| $\log(\lambda_1)$ | $[-3, \ \ 0]$ |
| $\log(\lambda_2)$ | $[-3, \ \ 0]$ |
| $\log(\lambda_3)$ | $[-1, \ \ 1]$ |
| $\eta_{noise}$ | $[0.3, 1.8]$ |

## D.2 MODEL SELECTION

During training, a policy is evaluated on a set of 10 different episodes every 100 learning steps. At the end of the training, the model at the best evaluation iteration is saved as the best version of the policy for this training, and is re-evaluated on 100 different episodes to have a better assessment of its final performance. The performance of a hyper-parameter configuration is defined as the average performance (across seeds) of the policies learned using this set of hyper-parameter values.

### D.3 SELECTED HYPER-PARAMETERS

Tables 2, 3, 4, and 5 shows the best hyper-parameters found by the random searches for each of the environments and each of the algorithms.

Table 2: Best found hyper-parameters for the SPREAD environment

| HYPER-PARAMETER | DDPG | MADDPG | MADDPG+SHARING | MADDPG+TEAMREG | MADDPG+COACHREG |
|---|---|---|---|---|---|
| $\alpha_\theta$ | $5.3 * 10^{-5}$ | $2.1 * 10^{-5}$ | $9.0 * 10^{-4}$ | $2.5 * 10^{-5}$ | $1.2 * 10^{-5}$ |
| $\omega_\phi$ | 53 | 79 | 0.71 | 42 | 82 |
| $\tau$ | 0.05 | 0.083 | 0.076 | 0.098 | 0.0077 |
| $\lambda_1$ | - | - | - | 0.054 | 0.13 |
| $\lambda_2$ | - | - | - | 0.29 | 0.24 |
| $\lambda_3$ | - | - | - | - | 8.4 |
| $\eta_{noise}$ | 1.0 | 0.5 | 0.7 | 1.2 | 1.2 |

Table 3: Best found hyper-parameters for the BOUNCE environment

| HYPER-PARAMETER | DDPG | MADDPG | MADDPG+SHARING | MADDPG+TEAMREG | MADDPG+COACHREG |
|---|---|---|---|---|---|
| $\alpha_\theta$ | $8.1 * 10^{-4}$ | $3.8 * 10^{-5}$ | $1.2 * 10^{-4}$ | $1.3 * 10^{-5}$ | $6.8 * 10^{-5}$ |
| $\omega_\phi$ | 2.4 | 87 | 0.47 | 85 | 9.4 |
| $\tau$ | 0.089 | 0.016 | 0.06 | 0.055 | 0.02 |
| $\lambda_1$ | - | - | - | 0.06 | 0.0066 |
| $\lambda_2$ | - | - | - | 0.0026 | 0.23 |
| $\lambda_3$ | - | - | - | - | 0.34 |
| $\eta_{noise}$ | 1.2 | 0.9 | 1.2 | 1.0 | 1.1 |

Table 4: Best found hyper-parameters for the CHASE environment

| HYPER-PARAMETER | DDPG | MADDPG | MADDPG+SHARING | MADDPG+TEAMREG | MADDPG+COACHREG |
|---|---|---|---|---|---|
| $\alpha_\theta$ | $4.5 * 10^{-4}$ | $2.0 * 10^{-4}$ | $9.7 * 10^{-4}$ | $1.3 * 10^{-5}$ | $1.8 * 10^{-4}$ |
| $\omega_\phi$ | 32 | 64 | 0.79 | 85 | 90 |
| $\tau$ | 0.031 | 0.021 | 0.032 | 0.055 | 0.011 |
| $\lambda_1$ | - | - | - | 0.06 | 0.0069 |
| $\lambda_2$ | - | - | - | 0.0026 | 0.86 |
| $\lambda_3$ | - | - | - | - | 0.76 |
| $\eta_{noise}$ | 0.6 | 1.0 | 1.5 | 1.0 | 1.1 |

Table 5: Best found hyper-parameters for the COMPROMISE environment

| HYPER-PARAMETER | DDPG | MADDPG | MADDPG+SHARING | MADDPG+TEAMREG | MADDPG+COACHREG |
|---|---|---|---|---|---|
| $\alpha_\theta$ | $6.1 * 10^{-5}$ | $3.1 * 10^{-4}$ | $6.2 * 10^{-4}$ | $1.5 * 10^{-5}$ | $3.4 * 10^{-4}$ |
| $\omega_\phi$ | 1.7 | 0.94 | 0.58 | 90 | 29 |
| $\tau$ | 0.065 | 0.045 | 0.007 | 0.02 | 0.0037 |
| $\lambda_1$ | - | - | - | 0.0013 | 0.65 |
| $\lambda_2$ | - | - | - | 0.56 | 0.5 |
| $\lambda_3$ | - | - | - | - | 1.3 |
| $\eta_{noise}$ | 1.1 | 0.7 | 1.3 | 1.6 | 1.6 |

## D.4   SELECTED HYPER-PARAMETERS (ABLATIONS)

Tables 6, 7, 8, and 9 shows the best hyper-parameters found by the random searches for each of the environments and each of the ablated algorithms.

Table 6: Best found hyper-parameters for the SPREAD environment

| HYPER-PARAMETER | MADDPG+AGENT MODELLING | MADDPG+POLICY MASK |
|---|---|---|
| $\alpha_\theta$ | $1.3 * 10^{-5}$ | $6.8 * 10^{-5}$ |
| $\omega_\phi$ | 85 | 9.4 |
| $\tau$ | 0.055 | 0.02 |
| $\lambda_1$ | 0.06 | 0 |
| $\lambda_2$ | 0 | 0 |
| $\lambda_3$ | - | 0 |
| $\eta_{noise}$ | 1.0 | 1.1 |

Table 7: Best found hyper-parameters for the BOUNCE environment

| HYPER-PARAMETER | MADDPG+AGENT MODELLING | MADDPG+POLICY MASK |
|---|---|---|
| $\alpha_\theta$ | $1.3 * 10^{-5}$ | $2.5 * 10^{-4}$ |
| $\omega_\phi$ | 85 | 0.52 |
| $\tau$ | 0.055 | 0.0077 |
| $\lambda_1$ | 0.06 | 0 |
| $\lambda_2$ | 0 | 0 |
| $\lambda_3$ | - | 0 |
| $\eta_{noise}$ | 1.0 | 1.3 |

Table 8: Best found hyper-parameters for the CHASE environment

| HYPER-PARAMETER | MADDPG+AGENT MODELLING | MADDPG+POLICY MASK |
|---|---|---|
| $\alpha_\theta$ | $2.5 * 10^{-5}$ | $6.8 * 10^{-5}$ |
| $\omega_\phi$ | 42 | 9.4 |
| $\tau$ | 0.098 | 0.02 |
| $\lambda_1$ | 0.054 | 0 |
| $\lambda_2$ | 0 | 0 |
| $\lambda_3$ | - | 0 |
| $\eta_{noise}$ | 1.2 | 1.1 |

Table 9: Best found hyper-parameters for the COMPROMISE environment

| HYPER-PARAMETER | MADDPG+AGENT MODELLING | MADDPG+POLICY MASK |
|---|---|---|
| $\alpha_\theta$ | $1.2 * 10^{-4}$ | $2.5 * 10^{-4}$ |
| $\omega_\phi$ | 0.71 | 0.52 |
| $\tau$ | 0.0051 | 0.0077 |
| $\lambda_1$ | 0.0075 | 0 |
| $\lambda_2$ | 0 | 0 |
| $\lambda_3$ | - | 0 |
| $\eta_{noise}$ | 1.8 | 1.3 |

### D.5 HYPER-PARAMETER SEARCH RESULTS

The performance of each parameter configuration is reported in Figure 10 yielding the performance distribution across hyper-parameters configurations for each algorithm on each task. The same distributions are depicted in Figure 11 using box-and-whisker plot. It can be seen that TeamReg and CoachReg both boost the performance of the third quartile, suggesting an increase in the robustness across hyper-parameter.

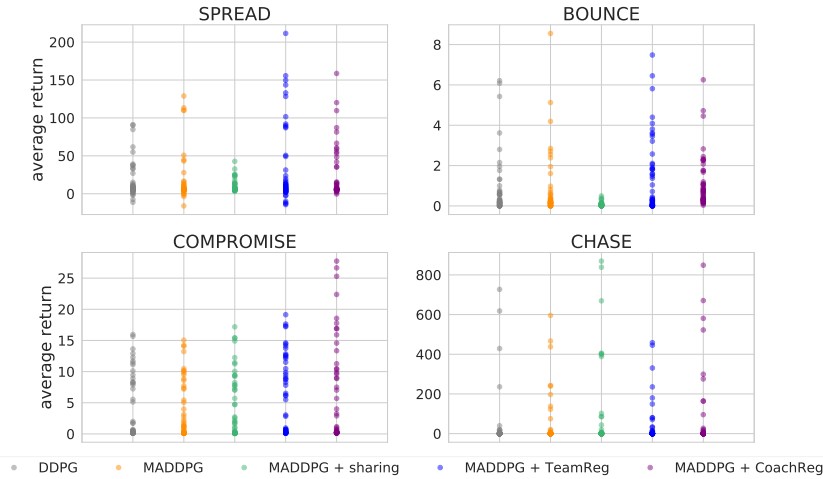

Figure 10: Hyper-parameter tuning results for all algorithms. There is one distribution per *(algorithm, environment)* pair, each one formed of 50 points (hyper-parameter configuration samples). Each point represents the best model performance averaged over 100 evaluation episodes and averaged over the 3 training seeds for one sampled hyper-parameters configuration (total of 300 performance values per sampled configuration).

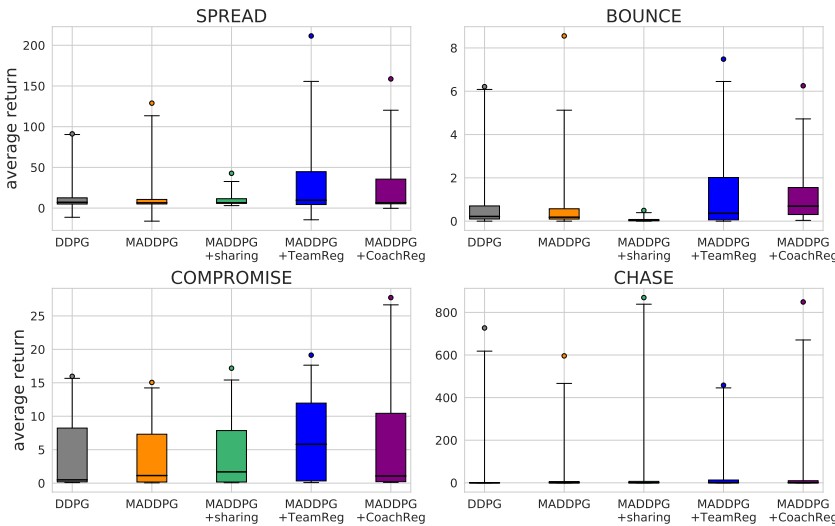

Figure 11: Summarized performance distributions of the sampled hyper-parameters configurations for each *(algorithm, environment)* pair. The box-plots divide in quartiles the 49 lower-performing configurations for each distribution while the score of the best-performing configuration is highlighted above the box-plots by a single dot.

### D.6 HYPER-PARAMETER SEARCH RESULTS (ABLATIONS)

From Figure 13 it seems that the "policy mask" or the "agent modelling" additions respectively provide nearly the same robustness boosts as CoachReg and TeamReg.

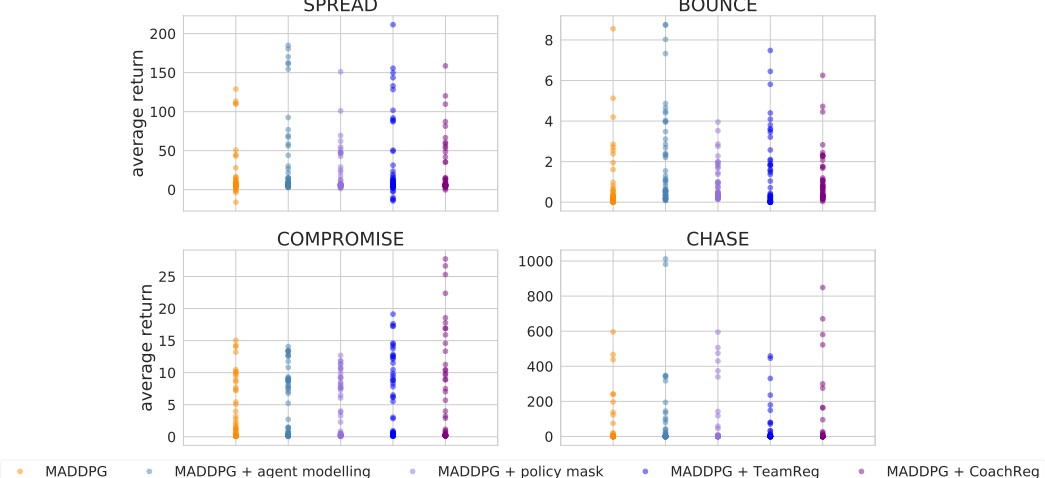

Figure 12: Hyper-parameter tuning results for ablated algorithms compared to their full approach counterparts and MADDPG. There is one distribution per *(algorithm, environment)* pair, each one formed of 50 points (hyper-parameter configuration sample). Each point represents the best model performance averaged over 100 evaluation episodes and averaged over the 3 training seeds for one sampled hyper-parameters configuration (total of 300 performance values per sampled configuration).

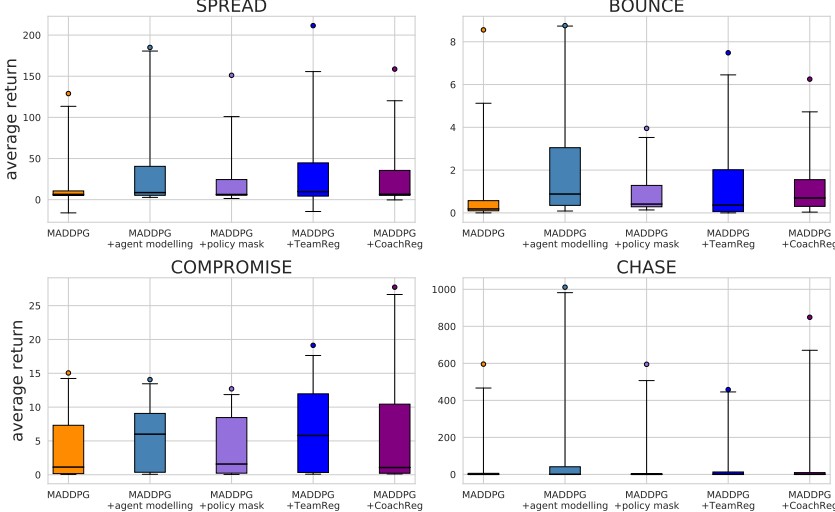

Figure 13: Summarized performance distributions of the sampled hyper-parameters configurations for each *(ablated algorithm, environment)* pair. The box-plots divide in quartiles the 49 lower-performing configurations for each distribution while the score of the best-performing configuration is highlighted above the box-plots by a single dot.

# E    THE EFFECTS OF ENFORCING PREDICTABILITY (ADDITIONAL RESULTS)

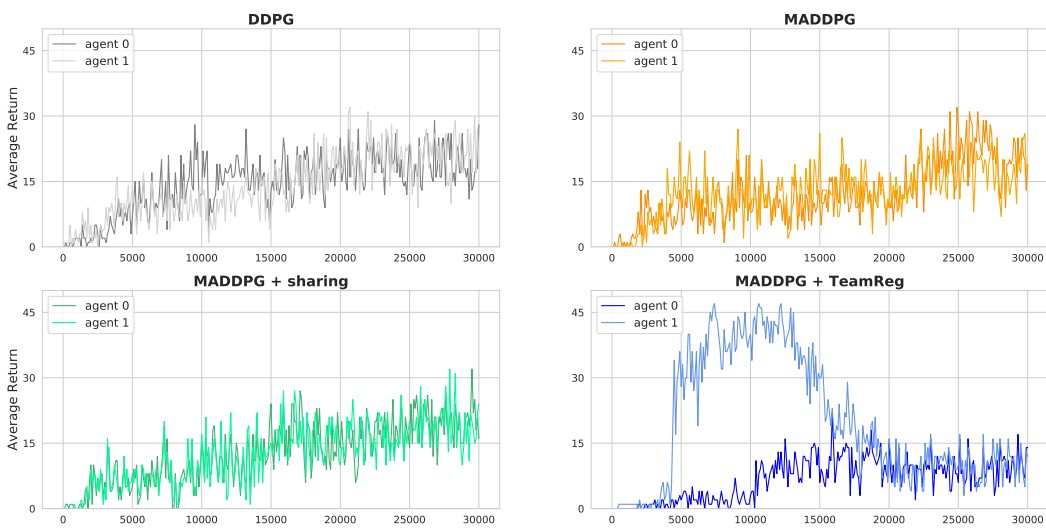

Figure 14: Learning curves for TeamReg and the three baselines on COMPROMISE. We see that while both agents remain equally performant as they improve at the task for the baseline algorithms, TeamReg tends to make one agent much stronger than the other one. This domination is optimal as long as the other agent remains docile, as the dominant agent can gather much more reward than if it had to compromise. However, when the dominated agent finally picks up the task, the dominant agent that has learned a policy that does not compromise see its return dramatically go down and the mean over agents overall then remains lower than for the baselines.

# F ANALYSIS OF SUB-POLICY SELECTION (ADDITIONAL RESULTS)

## F.1 MASK DENSITIES

We depict on Figure 15 the mask distribution of each agent for each *(seed, environment)* experiment. Firstly, in most of the experiments, agents use at least 2 different masks. Secondly, for a given experiments, agents' distributions are very similar, suggesting that they are using the same masks in the same situations and that they are therefore synchronized. Finally, agents collapse more to using only one mask on CHASE, where they also display more dissimilarity between one another. This may explain why CHASE is the only task where CoachReg does not improve performance. Indeed, on CHASE, agents do not seem synchronized nor leveraging multiple sub-policies which are the priors to coordination behind CoachReg. In brief, we observe that CoachReg is less effective in enforcing those priors to coordination of CHASE, an environment where it does not boost nor harm performance.

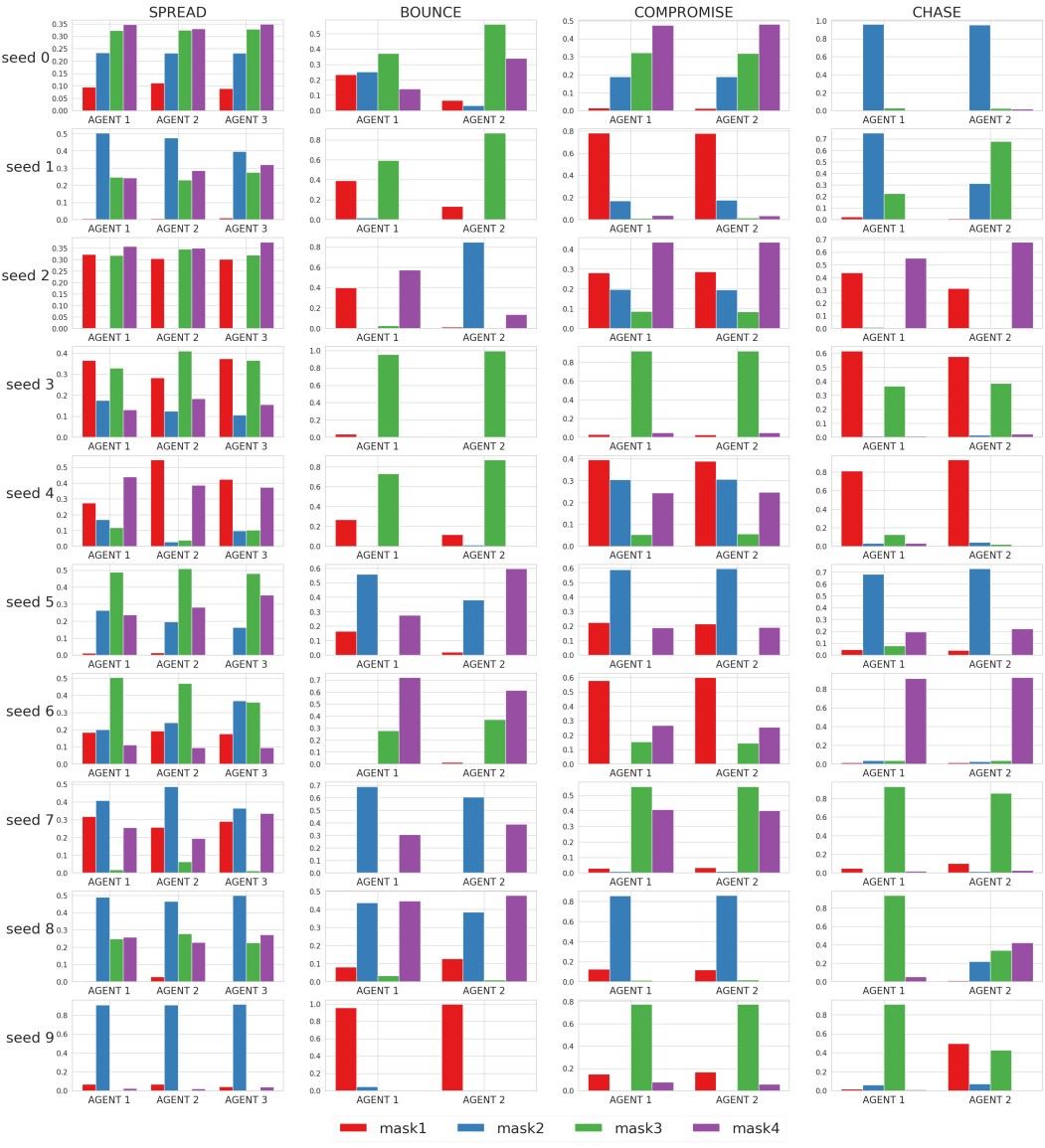

Figure 15: Agent's policy mask distributions. For each *(seed, environment)* we collected the masks of each agents on 100 episodes.

## F.2 EPISODES ROLL-OUTS

We render here some episodes roll-outs, the agents synchronously switch between policy masks during an episode. In addition, the whole group selects the same mask as the one that would have been suggested by the coach.

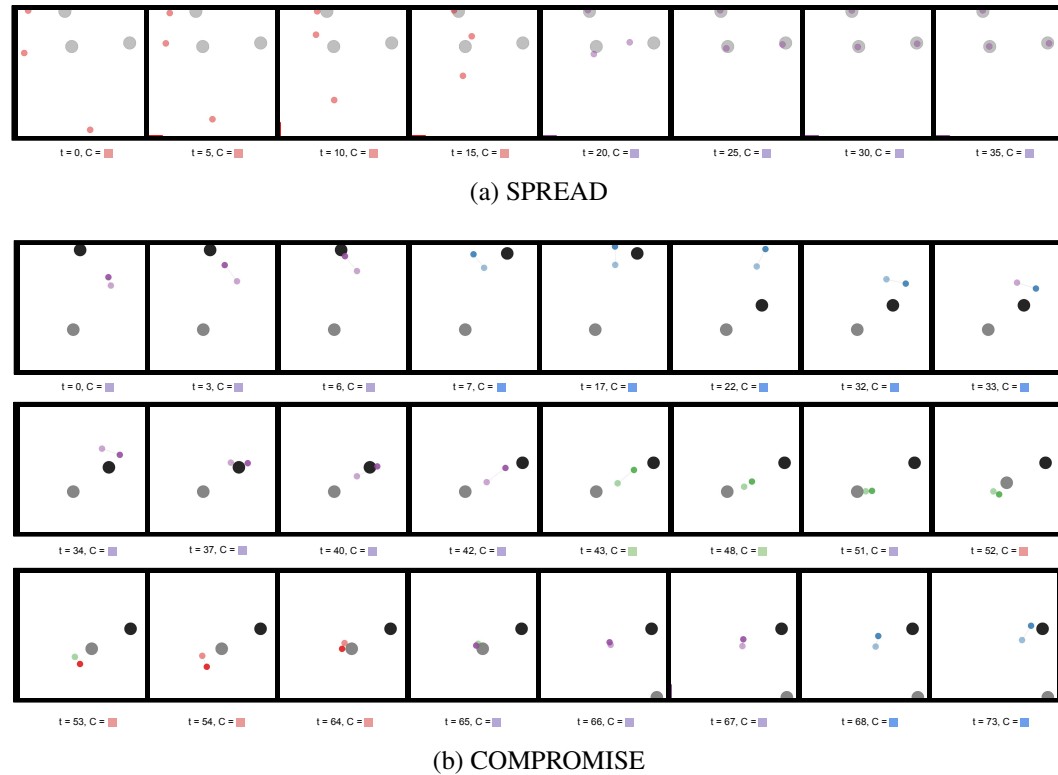

(a) SPREAD

(b) COMPROMISE

Figure 16: Visualization sequences on two different environments. An agent's color represent its current policy mask. For informative purposes the policy mask that the coach would have produced if these situations would have happened during training is displayed next to the frame's timestep. Agents synchronously switch between the available policy masks.

### F.3 MASK DIVERSITY AND SYNCHRONICITY (ABLATION)

As in Subsection 6.4 we report the mean entropy of the mask distribution and the mean Hamming proximity for the ablated "MADDPG + policy mask" and compare it to the full CoachReg. With "MADDPG + policy mask" agents are not incentivized to use the same masks. Therefore, in order to assess if they synchronously change policy masks, we computed, for each agent pair, seed and environment, the Hamming proximity for every possible masks equivalence (mask 3 of agent 1 corresponds to mask 0 of agent 2, etc.) and selected the equivalence that maximised the Hamming proximity between the two sequences.

We can observe that while "MADDPG + policy mask" agents display a more diverse mask usage, their selection is less synchronized than with CoachReg. This is easily understandable as the coach will tend to reduce diversity in order to have all the agents agree on a common mask, on the other hand this agreement enables the agents to synchronize their mask selection. To this regard, it should be noted that "MADDPG + policy mask" agents are more synchronized that agents independently sampling their masks from $k$-CUD, suggesting that, even in the absence of the coach, agents tend to synchronize their mask selection.

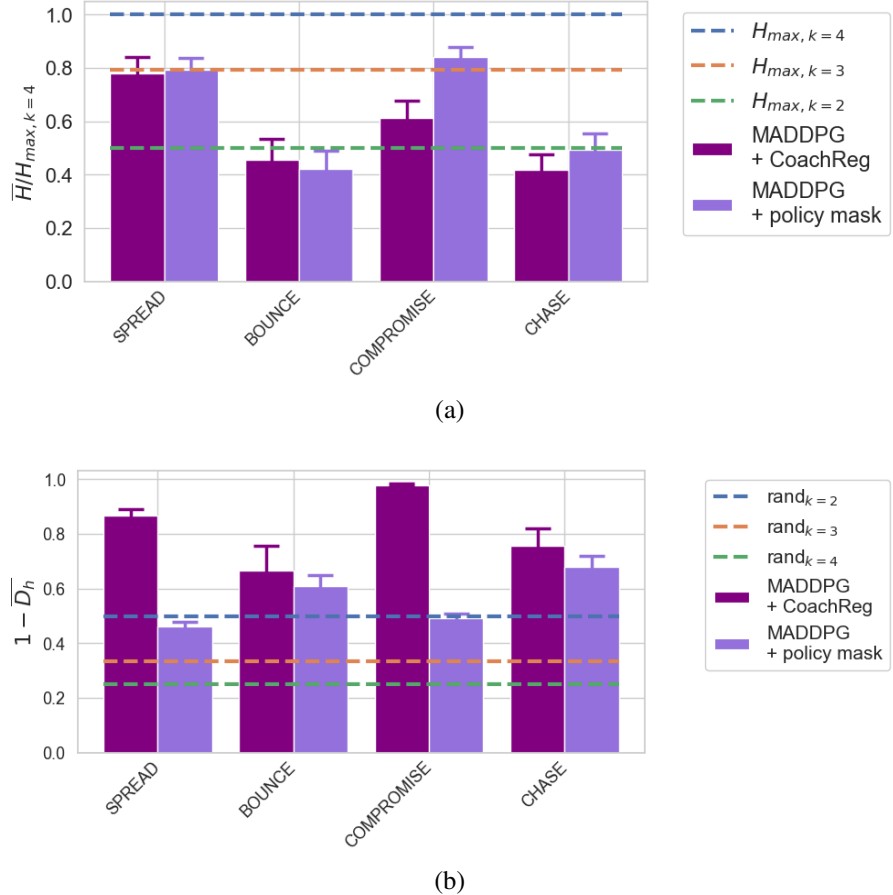

Figure 17: Entropy of the policy mask distributions for each task and method, averaged over agents and training seeds. $H_{max,k}$ is the entropy of a $k$-CUD. (b) Hamming Proximity between the policy mask sequence of each agent averaged across agent pairs and seeds. $rand_k$ stands for agents independently sampling their masks from $k$-CUD. Error bars are SE across seeds.

# G    ROBUSTNESS TO THE NUMBER OF AGENTS

We varied the number of agents present in the SPREAD task from three to six. For each algorithm we used the best performing hyper-parameter configuration from the hyper-parameter search performed on SPREAD with three agents and trained on ten different random seeds. Results are shown in Figure 18. As expected the task becomes more complicated when the number of agents increases and no algorithm succeeds at the task with six agents. This difficulty is likely to be exacerbated by the sparse reward setting. However, the proposed methods still outperform the baselines showing that they do not disproportionately suffer from the increased regularization pressure of additional agents.

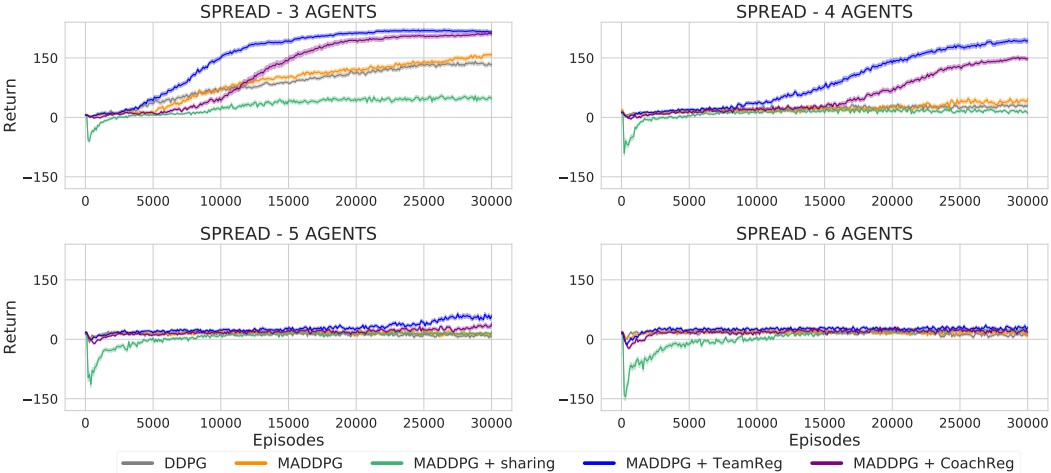

Figure 18: Learning curves (mean return over agents) for all algorithms on the SPREAD environment for varying number of agents. Solid lines are the mean and envelopes are the Standard Error (SE) across the 10 training seeds.

