# OpenReview forum: "Promoting Coordination through Policy Regularization in Multi-Agent Deep Reinforcement Learning"
_ICLR.cc/2020/Conference — Reject_

### Official Review · AnonReviewer2 · 2019-10-22
**Official Blind Review #2**

**Rating:** 8

**Review:**

This paper proposes two methods of biasing agents towards learning coordinated behaviours and evaluates both rigorously across two existing and two novel multi-agent domains of suitable complexity. The rigour of the empirical evaluation and its documentation is exemplary.

The related work section could be improved by including a section on the closely related work in the area of opponent modelling. This literature is touched on shortly afterwards alongside the related proposed method in Section 4.1 but only by a single citation. A more thorough review in Section 3 would improve the positioning of the paper.

From figures 4 and 5 it is interesting to observe that only parameter sharing performs significantly better than the existing baseline method MADDPG on the CHASE environment. In particular I think it would improve the discussion of the results to elaborate further on why the proposed methods do not improve, but also do not detrimentally affect the learning performance of agents in this environment.

In Section 6.2 the authors note "MADDPG + policy mask performs similarly or worse than MADDPG  on all but one environment and never outperforms the full CoachReg approach." I found it also interesting to note that in that one environment (SPREAD) adding a policy mask both outperformed MADDPG and was more sample efficient than CoachReg. Again what is it about this specific environment that causes the policy mask ablation to be sufficient, enabling it to match the asymptotic performance and learn quicker than the full CoachReg method proposed?

All references to papers both published and available on arxiv should cite the published version (e.g. Jacques et al. 2018 was published at ICML). Please revise all references if accepted.

Minor Comments:
1) Page 2 "two standards multi-agent tasks" -> standard
2) Page 7 "improve on MADDPG apart on the SPREAD environment" -> apart from on
3) Page 9 "each sampled configuration allows to empirically evaluate" -> allows us


**Experience Assessment:**

I have published in this field for several years.

**Review Assessment: Checking Correctness Of Derivations And Theory:**

I assessed the sensibility of the derivations and theory.

**Review Assessment: Checking Correctness Of Experiments:**

I carefully checked the experiments.

**Review Assessment: Thoroughness In Paper Reading:**

I read the paper thoroughly.

---

> ### Author Response · Authors · 2019-11-08
> **Response to Reviewer2 (1 of 1)**
>
> We wish to thank Reviewer2 for the feedback and for promoting discussions on some of the more intriguing results that fall from our experiments. We address you specific comments, questions and concerns, below.
>
> $\textbf{1.    “The related work section could be improved by including a section on the closely related work in the area}$$\textbf{of opponent modelling.”}$
>
> We agree: opponent modelling is closely related to our approach. We have reordered and added references to previous works in agent modelling (and opponent modelling) in our revised related work section, as per your suggestion. We also clarified how our use of agent modelling differs from this prior art.
>
>
> $\textbf{2.    “From figures 4 and 5 it is interesting to observe that only parameter sharing performs significantly better}$$\textbf{than the existing baseline method MADDPG on the CHASE environment. In particular I think}$$\textbf{it would improve the discussion of the results to elaborate further on why the proposed methods}$$\textbf{do not improve, but also do not detrimentally affect the learning performance of agents}$$\textbf{in this environment.“}$
>
> This is a very interesting comment. The discovery of coordinated behavior in this environment seems simpler than in our two proposed environments. Indeed, the presence of boundaries in the environment permits selfish and independent agent behaviors, where rushing independently towards the prey can perform surprisingly well given the right trajectory angles. This could explain why our methods, which aim to encourage the discovery of coordinated policies, provide no particular benefits on this task; however, as you mention, our mechanisms do not reduce performance.
>
> For TeamReg, we believe that any additional regularization pressure is negligible since the “rush towards the prey” behavior is already quite predictable.
>
> For CoachReg, the question is more complex (as illustrated in Figure 5): policy masks alone hurt performance whereas the full CoachReg approach does not. We suspect that the policy masks alone makes the sub-policies of the teammate alternate in an unpredictable manner, and so abrupt changes in its policy can perturb the prey (i.e., that is moved by repulsive forces) and prevent agents from trapping the prey when they are both close to it. Using CoachReg, sub-policies are jointly optimized and change synchronously, therefore the prey perturbation due to an agent changing its policy is immediately accounted for by the other agent (who is also changing its policy).
>
> Moreover, parameter sharing might be efficient on CHASE since this task is highly symmetric (one agent rushes from the left, another from the right, and they trap the prey) and thus we suspect that policy-specialisation may not be necessary to succeed for this task. For such problems where a shared policy is sufficient, it is to be expected that parameter sharing might prevail as it benefits from a more varied data collection process: transitions collected by both agents are used to train the same models.
>
>
> $\textbf{3.    “In Section 6.2 the authors note "MADDPG + policy mask performs similarly or worse than MADDPG}$$\textbf{on all but one environment and never outperforms the full CoachReg approach." I found it also interesting to note}$$\textbf{that in that one environment (SPREAD) adding a policy mask both outperformed MADDPG and was more}$$\textbf{sample efficient than CoachReg. Again what is it about this specific environment that causes the policy mask ablation}$$\textbf{to be sufficient, enabling it to match the asymptotic performance and learn quicker than the full CoachReg}$$\textbf{method proposed?”}$
>
> By inspecting the mask transitions sequences of CoachReg in that environment, we found that successful runs often result from sub-policy combinations such as (1) “rush towards landmark” and (2) “decelerate and stay on landmark” (see Figure 16, Appendix F2). The remaining difficulty lies in avoiding collisions with teammates. Synchronicity is not strictly necessary to put such a sequential strategy in place. At this point, sub-policy masks alone might be leveraged as a single-agent hierarchical component, easing the process of cutting down an episode into sequential segments. This may explain why — without the regularization pressure imposed by synchronicity — the “MADDPG+policy masks” ablation is able to match CoachReg final performance (while learning slightly faster).
>
>
> $\textbf{4.    Outdated (arXiv) citations and grammatical errors}$
>
> Thank you for pointing these issues: we have addressed them in our revised manuscript.
>
>
> $\textbf{5.    Conclusion}$
>
> We hope that our elaborations above satisfy your questions and concerns. We remain available to further discussions and/or clarifying any remaining ambiguities.

---

### Official Review · AnonReviewer3 · 2019-10-23
**Official Blind Review #3**

**Rating:** 3

**Review:**

Contribution:

This paper proposes two methods building upon MADDPG to encourage collaboration amongst decentralized MARL agents:
 - TeamReg: the agents are trained with an additional objectives of predicting other agents' actions, and make their own action predictable to them
 - CoachReg: the agents use a form of structured dropout, and at training time a central agent (the "coach") predicts the mask dropout mask that should be applied to all of the agents. An additional loss is provided so that the agent learn to mimic the coach's output from their own input so they can still apply the dropout at test time, when the coach is not available anymore.

These two methods are evaluated in 4 different MARL environments, and compared against their ablations and vanilla MADDPG.


Review:

The paper is well written and easy to follow. It generally motivates well the design choices, both intuitively and experimentally, using numerous ablations. It includes the analysis and explanations of the failure modes, which is valuable in my opinion.

The two main limitations of the work are the following:
 - Limited scale of the experiments. The majority of the experiments contain only 2 agents, and the last one merely contains 3. It is unclear whether the additional losses proposed in this work would still perform correctly with more agents, since the regularization pressure will increase.
 - No comparison to SOTA MARL methods. The only baseline method presented here is MADDPG, upon which this work is built. Recent work tend to show that it is easily outperformed, hence comparison to stronger baselines (QMIX, COMA, M3DDPG, ...) would be advisable to assess the quality of the policy found.


About the policy masks:
- what are the value chosen for K and C?
- Is there a reason why this particular form of dropout was chosen? Since it occurs after a fully-connected, it should be equivalent and more straight-forward to mask out C contiguous values.


Finally, it seems to me that the two methods presented here (TeamReg and CoachReg) are not mutually incompatible. Is there a reason why you didn't to apply both of them simultaneously?


**Experience Assessment:**

I have published one or two papers in this area.

**Review Assessment: Checking Correctness Of Derivations And Theory:**

I assessed the sensibility of the derivations and theory.

**Review Assessment: Checking Correctness Of Experiments:**

I assessed the sensibility of the experiments.

**Review Assessment: Thoroughness In Paper Reading:**

I read the paper at least twice and used my best judgement in assessing the paper.

---

> ### Author Response · Authors · 2019-11-08
> **Response to Reviewer3 (2 of 2)**
>
>
> $\textbf{3.    “About the policy masks: what are the value chosen for K and C?”}$
>
> Thank you for bringing this to our attention: we set $K$ to 4, meaning that agents can choose between 4 sub-policies. Since policies’ hidden layers have size 128, the corresponding value for $C$ is 32. We updated Appendix B in our revised manuscript to clarify this point.
>
>
> $\textbf{4.    “About the policy masks: is there a reason why this particular form of dropout was chosen? Since it occurs}$$\textbf{after a fully-connected, it should be equivalent and more straight-forward to mask out C contiguous values.”}$
>
> No, there is no particular reason. As you noted, since it is preceded and followed by fully-connected layers, the masking procedure that you suggest (masking an entire block of contiguous units of size $M/K$, where $M$ is the dimensionality of the layer and $K$ is the number of possible masks) is equivalent to the approach that we present in the paper (masking the $k$-th unit of every contiguous block of size $K$ on that layer). Since the layer does not encode any structural information, this operation can indeed be carried out in different ways, boiling down to a relatively minor implementation choice.
>
>
> $\textbf{5.    “it seems to me that the two methods presented here (TeamReg and CoachReg) are not mutually incompatible.}$$\textbf{Is there a reason why you didn't to apply both of them simultaneously?”}$
>
> You are right that the two approaches both aim at promoting coordination through different mechanisms and, so, could applied simultaneously. One would, however, need to account for the fact that — since each approach is independently implemented — their combination would require 5 additional hyperparameters on top of the base algorithm (as opposed to 2 hyperparameters for TeamReg or 3 for CoachReg).
>
> Since the hyperparameter space grows exponentially with the number of hyperparameters, tuning an MADDPG+TeamReg+CoachReg algorithm could become burdensome. Furthermore, analysing how these two secondary objectives interact would require many additional experiments. We leave such an exploration to future work and focus here on separately evaluating and analysing these two approaches.
>
>
> $\textbf{6.    Conclusion}$
>
> We thank Reviewer3 for very constructive feedback. We hope to have addressed the concerns that were raised in the review and that Reviewer3 will consider updating the rating to support the publication of this work. We remain available to further discussions and/or clarifying any remaining ambiguities.
>
>
> $\textbf{7.    References}$
>
> [1] Iqbal, S., & Sha, F. Actor-Attention-Critic for Multi-Agent Reinforcement Learning Supplementary Material.
>
> [2] Zheng, Lianmin, et al. "MAgent: A many-agent reinforcement learning platform for artificial collective intelligence." Thirty-Second AAAI Conference on Artificial Intelligence. 2018.
>
> [3] Suarez, J., Du, Y., Isola, P., & Mordatch, I. (2019). Neural MMO: A Massively Multiagent Game Environment for Training and Evaluating Intelligent Agents. arXiv preprint arXiv:1903.00784.
>
> [4] Foerster, J. N., Farquhar, G., Afouras, T., Nardelli, N., & Whiteson, S. (2018, April). Counterfactual multi-agent policy gradients. In Thirty-Second AAAI Conference on Artificial Intelligence.
>
> [5] Rashid, T., Samvelyan, M., De Witt, C. S., Farquhar, G., Foerster, J., & Whiteson, S. (2018). QMIX: monotonic value function factorisation for deep multi-agent reinforcement learning. arXiv preprint arXiv:1803.11485.
>
> [6] Hu, D., Jiang, X., Wei, X., & Wang, J. (2019, August). State Representation Learning for Minimax Deep Deterministic Policy Gradient. In International Conference on Knowledge Science, Engineering and Management (pp. 481-487). Springer, Cham.

---

> ### Author Response · Authors · 2019-11-08
> **Response to Reviewer3 (1 of 2)**
>
> We appreciate Reviewer3’s comments and interest in our work. The remainder of this posting will discuss and address the main concerns.
>
> $\textbf{1.    “Limited scale of the experiments. The majority of the experiments contain only 2 agents, and the last}$$\textbf{one merely contains 3. It is unclear whether the additional losses proposed in this work would still perform correctly}$$\textbf{with more agents, since the regularization pressure will increase.”}$
>
> Exploring the impact that increasing the number of agents has on our proposed methods is an interesting avenue: as such, we have included additional experiments in our revised manuscript (see Sections 6.6 and Appendix G). Here, we visualize learning curves of the methods on the SPREAD task, gradually doubling the number of agents from 3 to 6. We run each approach on 10 different random seeds using their most successful hyperparameter configuration found when tuning for the 3-agent version of the task.
>
> The results show that increasing the number of agents quickly increases the difficulty of solving the task. That being said, the results also show that — while none of the algorithms are able to learn the task in the given number of episodes for 5 or 6 agents — our two proposed methods are the only ones capable of performing the task in the 4-agent scenario.
>
> These additional experiments work towards addressing concerns regarding the scalability of our methods. We do not, however, claim that our methods suffice for scaling to a large number of agents, but we do believe that these results demonstrate that our auxiliary tasks are not the cause of the collapsed performance of these algorithms when more agents are involved. We hypothesise that value-function estimation might be the bottleneck here, as the input space of the centralized critics increases exponentially with the number of agents.
>
> Other works, such as Iqbal et al. [1], offer solutions for scalable multi-agent algorithms, e.g., the use of an attention model on the critics’ input to select only relevant information for an agent at any given timestep. That being said, other more complex dynamics might emerge as the number of agents increases, which is why massively multi-agent RL is in itself a distinct, open area of research with its own specialized tasks and challenges [2, 3]. Our work focuses instead on promoting coordination among a team of agents, which we believe to be non-trivial even in the context of small teams.
>
>
> $\textbf{2.    “No comparison to SOTA MARL methods. The only baseline method presented here is MADDPG, upon}$$\textbf{which this work is built. Recent work tend to show that it is easily outperformed, hence comparison to stronger}$$\textbf{baselines (QMIX, COMA, M3DDPG, ...) would be advisable to assess the quality of the policy found.”}$
>
> We understand this concern regarding the number of baselines, and we motivate our specific choices. COMA [4] has been published as concurrent work with MADDPG and can be seen as a multi-agent version of A2C (a vanilla actor-critic algorithm for discrete control). As such, it is not applicable to the continuous control setting. Similarly, QMIX [5] is also restricted to discrete control problems. M3DDPG [6] is indeed an interesting follow-up work on MADDPG, but is designed only for adversarial tasks.
>
> Our methods are designed for collaborative tasks and are compatible with a vast array of deep RL algorithms (as long as they use the centralised-training decentralised-execution framework). They can even be adapted to discrete control (CoachReg works out-of-the-box with discrete action spaces, whereas TeamReg would require us to define a new loss suited for comparing distributions, i.e., KL-divergence).
>
> We purposefully limited the scope of the types of problems we aim to address in our work and, while we agree that studying the behavior of the proposed methods on other base algorithms is an interesting avenue, we have chosen to leave this to future work. Our work instead focuses on the challenging domain of continuous control with sparse rewards, and so we compare against MADDPG which is the most popular approach for this class of tasks. Restricting ourselves to this (still interesting) subset of problems has allowed us to provide a more in-depth analysis — and thorough experimental suite — of our proposed methods (in particular: Sections 6.2, 6.3 6.4 and Appendices D5, D6, E and F).

---

### Official Review · AnonReviewer1 · 2019-10-24
**Official Blind Review #1**

**Rating:** 6

**Review:**

This paper proposed two approaches to encourage cooperation among multi-agents under the *centralized training decentralized execution* framework. The main contribution of this paper is that they propose to allow agents to predict the behavior of others and introduce this prediction loss into the RL learning objective. One approach is teammate-regularization where each agent predicts the behavior of all others (and therefore makes the total model complexity increasing quadratically with the number of agents), while the other is a centralized coach-regularization. The performance of these two approaches are compared with MADDPG and its two variants on 4 games. Experiments show that their approaches surpass a few baselines in certain game settings. However, the novelty of this algorithm is not strong, given that the similar idea of 'predicting the behavior of other agents' can be found in related work with discrete action spaces. Experimental results also show unstable advantages of their approaches.

Methodology:

Team-reg: One main difference between this approach and Jaques work: * Intrinsic social motivation via causal influence in multi-agent*  (or other MARL algorithms for discrete action spaces) is that in this work, each agent predict the action of other peers instead of the policy, since this algorithm is only based on DDPG with continuous action space. However, It seems that this idea is transferrable to discrete action spaces as well, so long as we change the MSE loss between actions to KL loss between policy over states. The novelty of this approach is not strong. Also, one hypothesis on which their work is based is that promoting agents’ predictability could foster such team structure and lead to more coordinated behaviors. Does predictability really improve cooperation? Could it be the other way around? Experiment results in the right-below penal (CHASE Game ) and left-above penal (SPREAD game) of Figure 5 actually strength my concern.

Coach-reg: Descriptions of the coach regularization approach is quite vague. I doubt the motivation and effectiveness of this approach. More questions are as follows:

* What is the role of the policy mask in deriving a better action? Is it just a dropout before the activation layer with a fixed $p$ proportion (which is related to the choice of $K$)? If so, this is like you first assume the policy network is somewhat overfitting, then alleviate this issue by letting the coach adjust which part to keep and which part to drop.  How would different $K$ values affect the performance?

* Does this framework really improve cooperation? Intuitively, all agents will finally reach a point where they agree on the same policy mask distribution (which is the one generated by the coach). How does the same policy mask lead to an improvement in cooperation? Empirical or theoretical explanations are strongly needed in the main results, not in the appendix section.

Experiments:

- All variants of MADDPG in the experiments are weak baselines, assuming that $MADDPG + sharing$ means all agents share the same policy and Q-function. However, since this algorithm only works for continuous action space, available relate work to compare with is quite limited. All environments only include two agents. Experiments with more than two agents should be implemented.
- Even for only 3 games (excluding the adversarial game setting), both branches failed to beat the *sharing* baseline for the CHASE game. For the ablation study, *agent-moduling* even works better than *TeamReg* for the CHASE game, so as the case when *policy mask* beats *CoachReg* in the SPREAD game (Figure 5). They seem to show that under these two proposed frameworks, the predictability of agents does not always encourage cooperation, at least for 2 out of 4 game settings mentioned in this section.
- This work makes a lot of effort on the hyper-parameter tuning part, which however does not provide a systematic solution to the tradeoff between improving predictability and maximizing reward.

**Experience Assessment:**

I do not know much about this area.

**Review Assessment: Checking Correctness Of Derivations And Theory:**

N/A

**Review Assessment: Checking Correctness Of Experiments:**

I carefully checked the experiments.

**Review Assessment: Thoroughness In Paper Reading:**

I read the paper thoroughly.

---

> ### Author Response · Authors · 2019-11-08
> **Response to Reviewer1 (6 of 6)**
>
>
> $\textbf{16.    Conclusion}$
>
> We thank Reviewer1 for taking the time to read our full response. We hope that our clarifications above, as well as the additional experiments and conclusions including the revised manuscript, address your concerns to the point where you are able to more clearly assess the totality of our contributions.
>
>
> $\textbf{17.    References}$
>
> [1] Iqbal, S., & Sha, F. Actor-Attention-Critic for Multi-Agent Reinforcement Learning Supplementary Material.
>
> [2] Zheng, Lianmin, et al. "MAgent: A many-agent reinforcement learning platform for artificial collective intelligence." Thirty-Second AAAI Conference on Artificial Intelligence. 2018.
>
> [3] Suarez, J., Du, Y., Isola, P., & Mordatch, I. (2019). Neural MMO: A Massively Multiagent Game Environment for Training and Evaluating Intelligent Agents. arXiv preprint arXiv:1903.00784.
>
> [4] Jaques, Natasha, Angeliki Lazaridou, Edward Hughes, Caglar Gulcehre, Pedro Ortega, Dj Strouse, Joel Z. Leibo, and Nando De Freitas. "Social Influence as Intrinsic Motivation for Multi-Agent Deep Reinforcement Learning." In International Conference on Machine Learning, pp. 3040-3049. 2019.
>
> [5] Li, Shihui, Yi Wu, Xinyue Cui, Honghua Dong, Fei Fang, and Stuart Russell. "Robust multi-agent reinforcement learning via minimax deep deterministic policy gradient." In AAAI Conference on Artificial Intelligence (AAAI). 2019.
>
> [6] Gupta, Jayesh K., Maxim Egorov, and Mykel Kochenderfer. "Cooperative multi-agent control using deep reinforcement learning." In International Conference on Autonomous Agents and Multiagent Systems, pp. 66-83. Springer, Cham, 2017.

---

> ### Author Response · Authors · 2019-11-08
> **Response to Reviewer1 (5 of 6)**
>
>
> $\textbf{12.    “All variants of MADDPG in the experiments are weak baselines”}$
>
> MADDPG is considered as the state-of-the-art in multi-agent RL with continuous action spaces and has been used as a robust baseline even in the most recent published works [5]. Moreover, MADDPG + Sharing (which is described in Section 6) makes use of parameter sharing across agents, a strategy that is very advisable for homogenous tasks that do not require specialisation [6]. Notably, it performs very well on one of the environment. Finally, we also include DDPG for completeness as it does not suffer from the exponential growth w.r.t. the number of agent of the inputs to the value function.
>
> It is important to note that methods we propose are compatible with most multi-agent deep RL methods and that the goal of our evaluation is to assess if they can be beneficial. To this end, we chose to extend MADDPG and to investigate how it interacts with our proposed methods.
>
>
> $\textbf{13.    “this algorithm only works for continuous action space”}$
>
> We present two distinct methods that aim at directing the policy search towards more coordinated group behaviors. Both of our methods can be used on top of any deep reinforcement learning algorithm that tackles the multi-agent problem from the Centralised Training Decentralised Execution perspective (see Section 1). While we have applied our two methods to continuous control tasks, there is no algorithmic or theoretical limitation to their application to discrete control tasks. One approach, CoachReg, works out-of-the-box with algorithms for discrete control since it only affects the internal representation of the actors. The other, TeamReg, can also be used for discrete control and would only need a different loss-function (as you mentioned: “it seems that this idea is transferable to discrete action spaces as well, so long as we change the MSE loss between actions to KL loss between policy”). We chose to focus our evaluations on continuous control tasks with sparse rewards because they are a challenging class of problems that have a lot of applications notably in robotics.
>
>
> $\textbf{14.    “This work makes a lot of effort on the hyper-parameter tuning part, which however does not provide}$$\textbf{a systematic solution to the tradeoff between improving predictability and maximizing reward.”}$
>
> We took great care when systematically tuning hyper-parameters across all the methods we demonstrate. As discussed in the paper, these searches on the generic hyper-parameters — such as learning rates or exploration noise — were designed so as to be fair across the compared algorithms. During the same search process, we also select the regularizing coefficients for TeamReg and CoachReg.
>
> Note that, in such a procedure, our methods are (purposefully) put at a disadvantage since we allocate the same fixed number of sampled configuration to each algorithm regardless of the fact that our methods require more hyper-parameters than every baseline. While tuning these regularization coefficients is not a systematic solution, it is a commonly adopted approach to finding a compromise between the auxiliary tasks and main objective.
>
>
> $\textbf{15.    “Empirical or theoretical explanations are strongly needed in the main results, not in the appendix section.”}$
>
> Our submitted manuscript is self-contained and does not mandate that the reader consult the appendices to understand and appreciate its contributions. Apart from the detailed content related to hyper-parameter tuning (which is very commonly relegated to supplementary material), all the data we present in the appendices are presented in some aggregated form in the main text. The appendix presents their raw form in an effort to satisfy those readers interested in the finer details.

---

> ### Author Response · Authors · 2019-11-08
> **Response to Reviewer1 (4 of 6)**
>
>
> $\textbf{9.    “Is it just a dropout before the activation layer with a fixed proportion $p$ (which is related to the choice of $K$)?}$
> $\textbf{If so, this is like you first assume the policy network is somewhat overfitting, then alleviate this issue by letting the coach}$$\textbf{adjust which part to keep and which part to drop.”}$
>
> The dropped proportion is indeed fixed, but it is not a stochastic dropout, i.e., dropping a unit with probability $p$ in order reduce overfitting to a training set. In our case, once the mask is selected the dropout is deterministic and structured: for a given mask we always drop the same units, therefore not targeting overfitting. On the contrary, this structured dropout ensures that only a subset of the policy network is trained for a given mask (related to the current world state) and therefore encourages sub-policies to overfit to the situations to which they apply.
>
>
> $\textbf{10.    “How would different K values affect the performance?”}$
>
> This is a very interesting question. A good choice of $K$ is related to the “modality” of a task. Indeed we see in Figure 9 (a) that agents tend to use mask as if they were equally alternating between $K=2$ masks in tasks with 2 agents (CHASE, COMPROMISE, BOUNCE) and $K=3$ for task with 3 agents (SPREAD). Further exploring this avenue is complementary to the goals of our work and left to future work.
>
>
> $\textbf{11.    “Experiments show that their approaches surpass a few baselines in certain game settings” [...]}$
> $\textbf{“Experimental results also show unstable advantages of their approaches”}$
>
> We tested our approaches comprehensively against three baselines on four tasks with sparse rewards. These tests are designed to assess the performance of our proposed algorithms on a variety of cooperative problems that require coordination in order to succeed. In addition to the baselines, we also compare our approaches to their ablated versions which are also competitive alternatives to the state-of-the-art. CoachReg always gets or matches the best asymptotic performance on all tasks except on CHASE where it is slightly outperformed by the parameter sharing baseline. Additionally, SPREAD is the only task where the use of policy mask alone is more sample efficient. TeamReg outperforms the baselines or performs on par, except on COMPROMISE, and outperforms “agent-modelling” in both sample efficiency and asymptotic performance, except on CHASE where it is less sample efficient. We rigorously analyse this failure mode of TeamReg on COMPROMISE in Section 6.3.
>
> We emphasize that the reliability of these results is ensured by our conservative evaluation methodology that allows the same fixed hyper-parameter search budget for the baselines as for our extensions, even if the latter has additional hyper-parameters. While it is difficult to find one singular method that outperforms all others on every single task in RL, we are confident that the experiments we present are conclusive and reproducible. They show that our approaches are beneficial on collaborative tasks, and often by a large margin.

---

> ### Author Response · Authors · 2019-11-08
> **Response to Reviewer1 (3 of 6)**
>
>
> $\textbf{6.    “Does predictability really improve cooperation? Could it be the other way around?”}$
>
> We do not directly refer to cooperation, but rather to coordination. We view the predictability of one agent’s behavior with respect to the other agents as a proxy of the degree of coordination between these agents. Therefore, we propose mechanisms to promote predictability in order to foster coordination, but in no way do we claim that predictability directly improves cooperation in general, only indirectly through the promotion of coordination.
>
> It seems intuitive that a cooperating team must be coordinated (how could two agents cooperate if they do not consider the agents they are cooperating with?). Yet, this point raised by Reviewer1 is interesting as this is not always the case: cooperation implies coordination, but not the other way around. For example, imagine that — in the COMPROMISE environment — agents are always moving in opposite directions. They would be perfectly coordinated as one could deduce the action of one agent exclusively from the action of the other agent. They would, however, not be able to solve that task that requires some degree of cooperation.
>
> We do not claim to be optimizing cooperation (in fact, it is hard to define cooperation and cast it as a learning objective). Rather, we claim that encouraging coordination along with the return maximization objective can ease the discovery of effective team strategies in tasks where cooperation is required to succeed. The results in the SPREAD and BOUNCE tasks (Figure 5) support our claim since TeamReg significantly outperforms its ablated version that does not promote predictability. These points are also supported by the more in-depth investigation we carried out in Section 6.3 (Figure 7).
>
>
> $\textbf{7.    “Descriptions of the coach regularization approach is quite vague. I doubt the motivation and effectiveness}$$\textbf{of this approach.”}$
>
> See Section 4.2: “In order to foster structured agents interactions, this method aims at teaching the agents to recognize different situations and synchronously select corresponding sub-behaviors.” The motivation for this approach comes from the fact that, in most team sports (e.g., soccer, handball or hockey), teams have two distinct functioning modes: either the team has possession of the ball and is playing offense, or it is playing defense. From individual observations, every player infers the team’s state and is able to synchronously switch between modes.
>
> We describe our method both in prose and in mathematical terms (in Sections 4.2.1 and 4.2.2). We are happy to revise the exposition given more precise critiques.
>
> Regarding the doubts expressed towards the effectiveness of our approach, we find the results in Figure 4 and Figure 5 to be convincing — they demonstrate that CoachReg is the overall best choice when viewing the ensemble of task performances.
>
>
> $\textbf{8.    “What is the role of the policy mask in deriving a better action?” [..] “How does the same policy mask}$$\textbf{lead to an improvement in cooperation?”}$
>
> The policy mask in itself is not meant to derive a better action, but rather to enforce that at a given time the actions of all the agents are derived from sub-policies that are jointly optimized for the identified situation (from which the mask is selected; please refer to Section 4.2). By this mechanism, the team can jointly react to a situation by changing their action selection mechanism.
>
> In the collective sports analogy, agents could have different sub-policies for offensive configurations than for defensive situations. They could also have different sub-policies when, e.g., an attack comes from the right than from the left.
>
> By having all team agents collectively select the same masks during training, we enforce that each sub-policy is constantly optimized with the corresponding teammates sub-policies. This facilitates the emergence of more elaborate joint behaviors since an agent’s sub-policies are fine-tuned to the coincident teammates’ sub-policies. Additionally, it should be easier to coordinate sub-policies used only in a subset of situations than to coordinate situation-agnostic policies.
>
> Yet, from Figure 5 we see that, except on SPREAD, the policy mask alone is not able to provide the desired effect and that synchronizing the policy selection through the use of a Coach is necessary to secure the benefits of the policy masking mechanism.

---

> ### Author Response · Authors · 2019-11-08
> **Response to Reviewer1 (2 of 6)**
>
>
> $\textbf{3.    “One approach is teammate-regularization where each agent predicts the behavior of all others (and therefore}$
> $\textbf{makes the total model complexity increasing quadratically with the number of agents)”}$
>
> The claim that team-regularization leads to a quadratic increase in “total model complexity” in the number of agents is mitigated by three important factors:
>
> (1) The critic is not affected by team-regularization, so our approach only increases complexity for the forward and backward propagation of the actor, which consists of roughly half of an agent’s computational load at training time;
>
> (2) Efficient design choices significantly impact real-world scalability and performance: we implement TeamReg by adding only additional heads to the pre-existing actor model (effectively sharing most parameters for the teammates’ action predictions with the agent’s action selection model). As such, only a small number of additional parameters need to be learned, and this does not lead to any significant negative performance impact compared to the underlying base algorithm.
>
> For a given agent, the number of parameters of the actor increases linearly with the number of agents (additional heads) whereas the critic model grows quadratically (since the observation size themselves depend on the number of agents). In the limit of increasing the number of agents, the proportion of added parameters by TeamReg compared to the increase in parameters of the centralised critic vanishes to zero.
> For example, on the SPREAD task, training 3 agents with TeamReg increases the number of parameters by about 1.25% (with similar computational complexity increase). With 100 agents, this increase is only of 0.48%; finally,
>
> (3) any additional heads are only used during training and can be safely removed at execution time, reducing the systems computational complexity to that of the base algorithm.
>
>
> $\textbf{4.    “the novelty of this algorithm is not strong, given that the similar idea of 'predicting the behavior of}$$\textbf{other agents' can be found in related work with discrete action spaces”}$
>
> We again have to assume here that this comment references TeamReg. We do not claim that predicting the behavior of teammates (often referred to as “agent-modelling”) as a component of TeamReg’s contribution. Instead, our contribution lies in the use of this objective to shape the behavior of teammates (see related work and Section 4.1 in the text).
>
> While prior art has recognized that multi-agent RL techniques should be exploring methods that are capable of predicting the behavior of other agents, there are many aspects to this problem that warrant greater scientific exploration. With TeamReg, for example, we extend the common agent-modelling objective so that it drives teammates to behave in a predictable manner. To our knowledge, no prior work in MARL applies such an explicit bias on the policy.
>
>
> $\textbf{5.    “One main difference between this approach and Jaques work: * Intrinsic social motivation via causal}$$\textbf{influence in multi-agent* (or other MARL algorithms for discrete action spaces) is that in this work, each agent predict}$$\textbf{the action of other peers instead of the policy”}$
>
> This concern is somewhat related to the previous one and is due, in part, to the misconception that “agent-modelling” is the main contribution of both Jaques et al.’s work [4] and our’s. This is incorrect. In both cases, the contribution lies in using the model of other agents rather as a means to compute a multi-agent objectives, be it social influence in Jaques et al.’s work or predicatibily in ours. Please refer to our discussion in the related work (Section 3) for more details.

---

> ### Author Response · Authors · 2019-11-08
> **Response to Reviewer1 (1 of 6)**
>
> We thank the Reviewer1 for the very detailed review. We will clarify potential misunderstandings, address direct questions, and validate our experimental methodology, below.
>
>
> $\textbf{1.    “The main contribution of this paper is that they propose to allow agents to predict the behavior of others”}$
>
> This statement seems to only refer to TeamReg, ignoring our CoachREg method and the novel cooperative tasks we proposed in our work (as summarized at the end of our introduction and throughout Section 4).
>
> Concretely, we propose and explore two ways to modify the classical RL objective. These modifications behave like auxiliary regularizing tasks and lead to greater agent coordination through different mechanisms. To our knowledge, CoachReg is the first multi-agent deep RL approach that aims at promoting coordination by enforcing synchronous sub-policy selection among agents. Similarly, TeamReg extends the common agent-modelling objective into a novel predictability constraint on the agents’ behavior.
>
>
> $\textbf{2.    “All environments only include two agents. Experiments with more than two agents should be implemented”}$
>
> Exploring the effect that increasing the number of agents has on our proposed methods is a very interesting avenue. To that extent, we have included additional experiments in our revised manuscript (see Sections 6.6 and Appendix G). In this new set of experiments, we visualize the learning curves of all methods on SPREAD when gradually doubling the number of agents from 3 to 6. We run each algorithm on 10 different random seeds with their most successful hyperparameter configuration found in the search for the 3-agents version of the task.
>
> The results show that increasing the number of agents very quickly makes the task more difficult to solve. That being said, the results also show that, while none of the algorithms are able to learn the task in the given number of episodes for 5 or 6 agents, our two proposed methods are the only ones that are able to perform in the 4 agents version.
>
> These additional experiments work towards addressing concerns regarding the scalability of our methods. Importantly, we do not claim that our methods are sufficient to be able to scale to a large number of agents, but we do believe that these results show that our auxiliary tasks are not the cause of the collapsed performance of these algorithms when more agents are involved. We hypothesise that the value-function estimation might be the bottleneck here, as the input space of the centralised critics increases exponentially with the number of agents. Other works, such as Iqbal et al. [1], offer solutions for scalable multi-agent algorithms, like the use of an attention model on the critics’ input to select only relevant information for an agent at a given timestep. However, other more complex dynamics might emerge as the number of agents increases, which is why mass multi-agent RL is in itself an open direction of research with specialized tasks and challenges [2, 3]. Our work focuses on the promotion of coordination among a team of agents, which we believe is a non-trivial goal even in the context of small teams.

---

### Author Response · Authors · 2019-09-26
**About code availability**

The code will be made public under license. As it is not possible to license the code without revealing the identity of the authors we had to turn off the link sharing. However, the code will be made available again upon publication of this work.

Thank you for your understanding.

The authors

---

### Author Response · Authors · 2019-11-08
**General Response to Reviewers**

We thank the reviewers for their detailed comments. We appreciate the positive feedback regarding our exposition, our methodology, and the breadth and depth of our experiments. We will respond to each reviewer and try to address all the concerns and questions that were raised. A few common remarks motivated modifications to the paper as well as the presentation of additional experiments and we have uploaded an updated version of our manuscript. The modifications are the following:

1) Clarified the distinction between our work and previous work on agent modelling in the Related Work section.

2) Added an experiment when increasing the number of agents in the SPREAD environment. This experiment is mentioned in the main text (Section 6.6) and a more detailed description as well as the resulting graphs have been added to the Appendix G.

3) Corrected a few grammatical errors, citations and notations.

---

### Decision · Program_Chairs · 2019-12-19

**Decision:**

Reject

**Comment:**

After reading the reviews and discussing this paper with the reviewers, I believe that this paper is not quite ready for publication at this time. While there was some enthusiasm from the reviewers about the paper, there were also major concerns raised about the comparisons and experimental evaluation, as well as some concerns about novelty. The major concerns about experimental evaluation center around the experiments being restricted to continuous action settings where there is a limited set of baselines (see R3). While I see the authors' point that the method is not restricted to this setting, showing more experiments with more baselines would be important: the demonstrated experiments do strike me as somewhat simplistic, and the standardized comparisons are limited.

This might not by itself be that large of an issue, if it wasn't for the other problem: the contribution strikes me as somewhat ad-hoc. While I can see the intuition behind why these two auxiliary objectives might work well, since there is only intuition, then the burden in terms of showing that this is a good idea falls entirely on the experiments. And this is where in my opinion the work comes up short: if we are going to judge the efficacy of the method entirely on the experimental evaluation without any theoretical motivation, then the experimental evaluation does not seem to me to be sufficient.

This issue could be addressed either with more extensive and complete experiments and comparisons, or a more convincing conceptual or theoretical argument explaining why we should expect these two particular auxiliary objectives to make a big difference.